# FACE: Faithful Automatic Concept Extraction

**Dipkamal Bhusal**
Rochester Institute of Technology
Rochester, NY
db1702@rit.edu

**Michael Clifford**
Toyota InfoTech Labs
Mountain View, CA
michael.clifford@toyota.com

**Sara Rampazzi**
University of Florida
Gainesville, FL
srampazzi@ufl.edu

**Nidhi Rastogi**
Rochester Institute of Technology
Rochester, NY
nxrvse@rit.edu

## Abstract

Interpreting deep neural networks through concept-based explanations offers a bridge between low-level features and high-level human-understandable semantics. However, existing automatic concept discovery methods often fail to align these extracted concepts with the model's true decision-making process, thereby compromising explanation *faithfulness*. In this work, we propose **FACE** (Faithful Automatic Concept Extraction), a novel framework that augments Non-negative Matrix Factorization (NMF) with a Kullback-Leibler (KL) divergence regularization term to ensure alignment between the model's original and concept-based predictions. Unlike prior methods that operate solely on encoder activations, FACE incorporates classifier supervision during concept learning, enforcing predictive consistency and enabling faithful explanations. We provide theoretical guarantees showing that minimizing the KL divergence bounds the deviation in predictive distributions, thereby promoting faithful local linearity in the learned concept space. Systematic evaluations on ImageNet, COCO, and CelebA datasets demonstrate that FACE outperforms existing methods across faithfulness and sparsity metrics.

## 1 Introduction

Interpreting the decisions made by deep learning models is essential for understanding their behavior, diagnosing biases, correcting failed models, and fostering user trust. Among a wide array of explainable AI (XAI) methods proposed in recent years, feature attribution methods dominate the current practice [3, 10, 31, 30, 35]. These methods assign importance scores to individual input features (e.g., pixels in images), based on their influence on the model's output. However, such fine-grained attributions fail to deliver semantic interpretability as they do not clarify how specific high-level concepts drive a model's decision [16]. For instance, in animal image classifiers, assigning importance to isolated pixels reveals little about the body parts or visual cues the model actually uses, making it difficult for users to interpret decisions or assess failure modes meaningfully [6, 23, 29].

Concept-based explanation methods have recently emerged to address the interpretability shortcoming of feature attribution methods [8, 12, 16, 34, 36]. These methods explain model predictions using human-interpretable concepts (e.g., texture, part, color), enabling a more intuitive understanding of the model's reasoning process. For instance, rather than attributing importance to specific pixels, a concept-based explanation might reveal that the model relies on the presence of 'fur' or 'ears' in an animal image classifier (see Figure 1).

39th Conference on Neural Information Processing Systems (NeurIPS 2025).

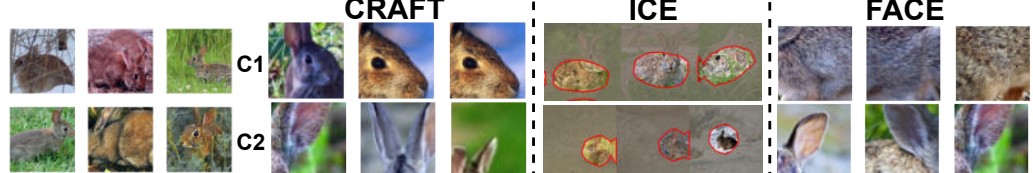

Figure 1: Comparing concepts extracted by CRAFT [8], ICE [36], and FACE from rabbit images classified by ResNet-34 [14]. C1 and C2 correspond to the top two Sobol-importance concepts for each method. FACE achieves higher faithfulness compared to CRAFT and ICE (discussed in Section 4.2), demonstrating that FACE's extracted concepts better align with the model's true reasoning.

However, existing concept-based methods face significant limitations. Early methods like TCAV [16] require a manually-labeled dataset for extracting concepts, greatly limiting scalability and practical applicability. Unsupervised methods such as ACE [12], ICE [36], and CRAFT [8] automate the concept discovery using unsupervised clustering or matrix factorization of encoder activations. Non-negative matrix factorization (NMF) [18] based approaches like CRAFT [8], in particular, have been shown to produce spatially coherent and interpretable concepts [9]. However, all these methods focus solely on reconstructing the latent representation and *do not* account for the behavior of the model's downstream classifier. As a result, the explanations may appear interpretable and yet misinterpret the original model's true reasoning, resulting in misleading insights.

Figure 1 illustrates this misalignment: CRAFT [8] attributes the prediction of a rabbit to the topmost concept (C1) 'head', while our method, FACE, highlights 'fur (body)' as the most important concept (C1). Although 'head' may be more intuitive to humans, it is fundamentally flawed to assume that the neural network relies on human-aligned features [11].

Thus, faithfulness- how well the explanations align with the actual decision-making process of the model- is a critical and often under-examined requirement in concept-based methods. Moreover, as shown in the identifiability result of Locatello et al. [21], discovering meaningful latent concepts in an unsupervised setting is fundamentally ill-posed without suitable inductive biases. In concept discovery, enforcing consistency between concept-based representations and the model's predictions provides such an inductive bias, helping to disambiguate useful representations from spurious ones.

**Our solution:** To address these challenges, we propose **FACE (Faithful Automatic Concept Extraction)**, a novel NMF-based concept discovery framework that enforces alignment between concept-based activations and model predictions. FACE augments standard NMF by introducing a *KL divergence constraint* that minimizes the discrepancy between the model's predictions on the original and reconstructed activations. This additional regularization ensures that the discovered concepts do not merely compress high-variance directions in latent space, but also closely align with the model's true predictive reasoning. By supervising the factorization process using the model's own predictions, FACE yields explanations that are both semantically interpretable and faithful–capturing the actual features used by the model in making its predictions. We provide theoretical justification for our method in Section 3.3 and empirically validate its effectiveness on ImageNet [7], COCO [19] and CelebA [20] in Section 4.

Our code is available at `https://github.com/dipkamal/FACE`.

## 2 Related work

Kim et al. [16] introduced Testing with Concept Activation Vectors (TCAV) to explain neural network predictions using human-defined concepts by training a linear classifier on activation vectors of specific concepts and random counterexamples. However, TCAV relies on manually curated concept datasets, limiting scalability and introducing human bias. ACE [12] addresses this by segmenting images into superpixels (e.g., via SLIC [1]) and clustering segment-level activations to discover concepts automatically. While ACE removes manual concept curation, it often introduces artifacts in concepts and may not align with model's actual decision-making. ICE [36] replaces clustering with Non-negative matrix factorization (NMF) [18], offering cleaner concepts but operates at the

level of convolutional kernels, discovering localized concepts only. CRAFT [8] improves on ICE by introducing a recursive factorization pipeline that decomposes high-level concepts into sub-concepts. It applies NMF to cropped image activations and proposes a concept scoring method based on Sobol indices [32]. While CRAFT improves interpretability and introduces a principled way to score concept relevance, it still operates purely on the encoder activations, and the NMF decomposition does not enforce predictive consistency through the classifier. Other recent works explore instance-level concepts using SAM [34] or study concept composition [33]. In parallel to these reconstruction-based methods, Achtibat et al. [2] proposed Concept Relevance Propagation (CRP) and Relevance Maximization (RelMax). Unlike ACE, ICE, or CRAFT, RelMax does not decompose activations but instead selects representative training examples for naturally emerging neurons or features by maximizing their relevance during inference. In contrast to these works, our proposed method, FACE, introduces a KL-regularized NMF framework that explicitly aligns discovered concepts with the classifier's predictive behavior. By supervising concept extraction with model predictions, FACE ensures that discovered concepts remain faithful to the model, an aspect overlooked in prior NMF-based approaches.

## 3 Methodology

### 3.1 Setup

Consider a standard supervised learning setup where a classifier $f : \mathcal{X} \to \mathcal{Y}$ maps inputs from an input space $\mathcal{X} \subseteq \mathbb{R}^d$ to output predictions in $\mathcal{Y} \subseteq \mathbb{R}^c$. Without loss of generality, we assume that $f$ can be decomposed into two components: an encoder $g : \mathcal{X} \to \mathcal{G}$ and a classifier head $h : \mathcal{G} \to \mathcal{Y}$, where $\mathcal{G} \subseteq \mathbb{R}^p$ is an intermediate latent space. The encoder $g$ maps inputs to the high-dimensional latent representation, while $h$ maps the latent representations to the output logits, or class-scores. Consequently, $f(\mathbf{x}) = h(g(\mathbf{x}))$. Given a dataset of $n$ input samples $\mathbf{X} = [\mathbf{x}_1, \dots, \mathbf{x}_n] \in \mathbb{R}^{n \times d}$, we denote their corresponding latent representations as $\mathbf{A} = g(\mathbf{X}) \in \mathbb{R}^{n \times p}$.[1]

Automatic concept discovery methods operate on these latent representations $\mathbf{A}$, typically for a set of inputs from the same class. To identify human-interpretable concepts, methods such as clustering [12], or Non-negative matrix factorization (NMF) [8, 36] is applied on $\mathbf{A}$ to discover a set of $k$ interpretable concept vectors. Here, the value of $k$ is typically pre-specified as the number of clusters for clustering-based approaches or the rank of matrix decomposition for matrix factorization based approaches. Once the concepts are discovered, their importance to the prediction is quantified using sensitivity-based methods such as TCAV [16] or variance decomposition methods such as Sobol indices [8].

In this work, we adopt the NMF-based pipeline with Sobol-based concept importance [8] (see Appendix M for a discussion on Sobol index) due to its demonstrated effectiveness in extracting interpretable and spatially coherent concepts [8, 9].

### 3.2 Desiderata of explanations

The goal of explanations is to help end-users understand how a neural network makes a decision. This information needs to be semantically interpretable so that it's easily understood by the users, however, it should also remain faithful to the underlying model. Below, we outline two important desiderata for explanations:

**High Faithfulness:** A critical requirement of any explanation method is that it accurately reflects the model's decision-making process. An explanation is considered faithful if the features or concepts it highlights are indeed those that the model uses to make its predictions. However, human intuitions are often poor proxies for model behavior [11], which motivates the use of evaluation metrics grounded in model performance. We adopt two perturbation-based metrics from feature attribution literature [25, 27]: *Concept Deletion (C-Del)* measures the drop in model accuracy as the most important concepts are progressively removed from the latent representation. A faithful method will

---

[1]While FACE is formulated using the penultimate layer to capture high-level, semantically coherent concepts, it is not restricted to this layer. For any intermediate layer $l$, we simply have to define $g_{\leq \ell}$ as the network up to that layer and $h_{> \ell}$ as the remaining layers. We focus on the penultimate layer to align with prior works and to target class-level concepts—empirically, earlier layers correspond to texture- or edge-like features, whereas deeper layers capture localized, semantically meaningful parts (for details, see [24]).

lead to a steep accuracy drop when key concepts are deleted. We compute *C-Del* as the area over the accuracy curve where higher scores indicate stronger reliance on the removed concepts, and thus more faithful explanations. The second metric, *Concept Insertion (C-Ins)* measures how rapidly accuracy is recovered when important concepts are reintroduced into a blank representation. A high *C-Ins* score implies that the explanation provides concepts used by the model in prediction.

**Low Complexity:** Semantic interpretability is also contingent on the cognitive simplicity of the explanation. Sparse and focused explanations are easier for users to comprehend and reason about. We measure this property using the sparsity metric of Gini index [5] (*C-Gini*) computed over the vector of concept importance scores. A higher Gini index corresponds to less complex explanations, where a few dominant concepts explain most of the prediction.

See Appendix L.2 for discussion on evaluation metrics of faithfulness and complexity.

### 3.2.1 Problem Formulation

Concept-based explanation via Non-negative matrix factorization (NMF) seeks to represent a non-negative neural activation as a composition of a small set of interpretable basis vectors. Given an activation matrix $\mathbf{A} \in \mathbb{R}_+^{n \times p}$ extracted from a neural network's encoder (e.g., penultimate layer activations), classical NMF factorizes it into a non-negative dictionary matrix $\mathbf{W} \in \mathbb{R}_+^{p \times r}$ and coefficient matrix $\mathbf{U} \in \mathbb{R}_+^{n \times r}$, such that the original activation is approximated as $\mathbf{A} \approx \mathbf{U}\mathbf{W}^\top$. This is typically framed as the following optimization problem:

$$\min_{\mathbf{U} \geq 0, \, \mathbf{W} \geq 0} \frac{1}{2} \|\mathbf{A} - \mathbf{U}\mathbf{W}^\top\|_F^2 \tag{1}$$

where the Frobenius norm penalizes the reconstruction error between the original activations and their low-rank approximation. This formulation ensures that activations are approximately reconstructed using a sparse and additive combination of concept basis vectors.

However, minimizing only the reconstruction loss emphasizes preserving the activation geometry in the encoder space and does not guarantee alignment with the model's predictive behavior. As a result, the reconstructed activations $\mathbf{U}\mathbf{W}^\top$ may differ significantly in terms of downstream predictions through the classifier head $h$. This can yield concept explanations that appear meaningful to humans, yet fail to capture the actual features the model relies on. Moreover, classical NMF tends to prioritize directions of high variance in $\mathbf{A}$, which are not necessarily predictive or, class-discriminative. These factors undermine the *faithfulness* of the explanations, as the extracted concepts may not align with the model's decision-making process. We discuss the failure cases of standard NMF-based techniques in Section 3.4.

To address this limitation, we propose a faithfulness-aware variant of NMF that explicitly aligns the reconstructed activations with the model's predictive behavior. This is achieved by introducing a Kullback-Leibler (KL) divergence constraint between the classifier head prediction on $\mathbf{A}$ and $\mathbf{U}\mathbf{W}^\top$, resulting in the following objective:

$$\min_{\mathbf{U} \geq 0, \, \mathbf{W} \geq 0} \frac{1}{2} \|\mathbf{A} - \mathbf{U}\mathbf{W}^\top\|_F^2 + \lambda \cdot \mathrm{KL}(h(\mathbf{A}) \| h(\mathbf{U}\mathbf{W}^\top)) \tag{2}$$

Here, $h(\cdot)$ denotes the classifier head that maps activations to logits, and $\lambda > 0$ controls the trade-off between reconstruction fidelity and predictive alignment. KL divergence is computed over the softmax-normalized logits, enforcing consistency between the model's predictions on the original and concept-based representations. This modification ensures that the learned low-dimensional concept representation $\mathbf{U}$ retains both the structure of the encoder activations and the predictive semantics of the classifier, improving the faithfulness of the discovered concepts.

Unlike classical NMF, which can be optimized using multiplicative update rules [17], our formulation introduces a non-Euclidean KL term that depends on the downstream classifier, making multiplicative updates inapplicable. We optimize Eqn. 2 using projected gradient descent, alternately updating $\mathbf{U}$ and $\mathbf{W}$ while enforcing non-negativity constraints via projection. Convergence to a stationary point is guaranteed under mild conditions (see Appendix C). To improve convergence speed and stability, we initialize the factor matrices $\mathbf{U}$ and $\mathbf{W}$ using the Non-negative Double Singular Value Decomposition (NNDSVD) method [4] (see Appendix I.1).

Our formulation in Eqn. 2 can also be viewed through the lens of supervised dictionary learning [22], where the goal is to learn a dictionary $\mathbf{W}$ not only for reconstructing inputs but also for improving downstream task performance. In our case, rather than using explicit class labels, we impose supervision through model prediction alignment, ensuring that the learned concept representations remain faithful to the model's decision-making process.

### 3.3 Faithfulness Guarantee via KL Divergence Regularization

We provide a theoretical justification for incorporating a KL divergence regularization term between the classifier head predictions on the original and reconstructed activations. This constraint encourages the reconstructed activation space to preserve the model's predictive behavior, thereby promoting faithful concept extraction.

#### 3.3.1 Reconstruction *alone* is not faithful

Let $\mathbf{A} \in \mathbb{R}_+^{n \times p}$ be the activation matrix obtained from an encoder $g$, and let $\mathbf{A}' = \mathbf{U}\mathbf{W}^\top$ be its non-negative low-rank approximation obtained via non-negative matrix factorization (NMF). A first–order Taylor expansion of the fixed classifier head $h$ around $\mathbf{U}\mathbf{W}^\top$ gives

$$h(\mathbf{A}') - h(\mathbf{A}) \approx \nabla h(\mathbf{A}') \cdot (\mathbf{A} - \mathbf{A}').$$

where $\nabla h$ denotes the Jacobian of $h$ at $\mathbf{U}\mathbf{W}^\top$. This highlights that even when $\mathbf{A}$ is close to $\mathbf{U}\mathbf{W}^\top$ in activation space, the corresponding logits can differ significantly depending on $\nabla h$. Therefore, minimizing only the reconstruction error $\|\mathbf{A} - \mathbf{U}\mathbf{W}^\top\|_F^2$ is insufficient for ensuring predictive alignment. The following example makes this concrete:

Consider a linear classifier head followed by a temperature–scaled soft-max:

$$y = h(\mathbf{z}) = W_h \mathbf{z}, \qquad \sigma_\alpha(\mathbf{y}) = \sigma(\alpha\, \mathbf{y}),$$

where $W_h$ is fixed and $\alpha > 0$ fixed by training due to weight norm or batch-norm. Let $\mathbf{A}$ be the true activation and let $\mathbf{A}' = \mathbf{U}\mathbf{W}^\top$ with a tiny reconstruction error $\delta = \mathbf{A}' - \mathbf{A}$ so that $\|\delta\|_2 = \varepsilon_{\text{rec}} \ll 1$. The logit difference is $\Delta \mathbf{y} = h(\mathbf{A}') - h(\mathbf{A}) = W_h \delta$. and after scaling by $\alpha$, the predictive distribution is $\sigma_\alpha(\cdot)$. Although $\delta$ is small, the probabilities can be made *arbitrarily large* by the fixed gain $\alpha$ and the weight norm $W_h$. Hence, low Frobenius error on reconstruction vector offers no control over predictions.

This shows that reconstruction loss alone cannot guarantee faithfulness; one must constrain the predictive distribution itself. To address this, our formulation adds the KL divergence regularization between the softmax-normalized logits: $\lambda \cdot \text{KL}(\text{softmax}(h(\mathbf{A})) \| \text{softmax}(h(\mathbf{U}\mathbf{W}^\top)))$, which explicitly penalizes deviations in the predicted distributions.

Let $\mathbf{p} = \text{softmax}(h(\mathbf{A}))$ and $\mathbf{q} = \text{softmax}(h(\mathbf{A}'))$ be the predictive distributions before and after reconstruction. FACE minimises $L(\mathbf{U}, \mathbf{W}) = \frac{1}{2}\|\mathbf{A} - \mathbf{U}\mathbf{W}^\top\|_F^2 + \lambda \, \text{KL}(\mathbf{p}\|\mathbf{q})$. As a direct consequence of minimizing this KL divergence, we can bound the difference between the two predictive distributions. By Pinsker's inequality [26], constraining the KL divergence to be less than or equal to $\varepsilon$ guarantees that the total variation distance is bounded:

$$\|p - q\|_1 \leq \sqrt{2 \cdot \text{KL}(\cdot\|\cdot)} \leq \sqrt{2\varepsilon} \tag{3}$$

**Implication.** This result formally establishes that minimizing the KL divergence between predictive distributions directly constrains the deviation in model outputs caused by replacing the original activation $\mathbf{A}$ with its concept-based approximation $\mathbf{U}\mathbf{W}^\top$. In contrast, minimizing only the reconstruction loss $\|\mathbf{A} - \mathbf{U}\mathbf{W}^\top\|_F^2$ does not offer any such guarantee. Note that the value of $\varepsilon$ in Eqn. 3 is not a fixed constant but simply the empirical value of $\text{KL}(\mathbf{p}\|\mathbf{q})$. We do not assume a priori bound on $\varepsilon$ but our hyperparameter $\lambda$ from Eqn.2 controls how small $\varepsilon$ becomes. To make this explicit, we discuss $\varepsilon$ values over a range of $\lambda$ in Appendix E.

#### 3.3.2 KL divergence regularization *leads to* local linearity in the concept space

Interpretability methods often seek local linear approximations of complex neural networks that faithfully capture the model's behavior in a small neighborhood of interest [13]. In our formulation,

KL divergence regularization between the model predictions on the original and reconstructed activations encourages the local linearity of the classifier $h$ with respect to the concept representation $\mathbf{U}$.

Although the classifier head $h$ is typically a linear (e.g., a fully connected layer), the final model prediction involves a softmax applied to the logits $h(\mathbf{A})$, introducing nonlinearity. This means, even small changes in activations can still lead to disproportionate changes in predicted class probabilities. Consequently, minimizing only the reconstruction error $\|\mathbf{A} - \mathbf{U}\mathbf{W}^\top\|$ does not guarantee local linearity at the level of predicted distributions.

As discussed in Section 3.3.1, minimizing the KL divergence directly constrains deviations in predictive distributions, as shown by Pinsker's inequality [26]: for KL divergence bounded by $\varepsilon$, change in prediction is bounded by $\sqrt{\epsilon}$ thereby ensuring tight control over model prediction.

**Implication.** KL regularization ensures that the softmax operates in a regime where its nonlinearity is negligible, i.e., it behaves almost linearly around $\mathbf{U}\mathbf{W}^\top$. This promotes faithful, predictable changes in model output with respect to manipulations in the concept representation $\mathbf{U}$. Hence, the discovered concepts form a locally linear and interpretable basis for understanding model behavior.

### 3.4 Failure Case of Unconstrained NMF

To highlight the limitations of standard NMF-based concept discovery methods such as CRAFT [8] & ICE [36], we evaluate their ability to preserve model predictions on reconstructed activations. Specifically, we analyze classifier performance on reconstructed activations $\hat{\mathbf{A}} = \mathbf{U}\mathbf{W}^T$ from ResNet-34 [14] on ImageNet [7] and COCO [19]. We begin with a set of input images that are *correctly classified with 100% accuracy* by the original model, ensuring that any deviation in prediction is caused solely by the reconstruction process.

Figure 2a shows that while FACE consistently retains 100% top-1 accuracy across all classes, CRAFT and ICE suffer notable drops, e.g., only 40% accuracy for "Train" (CRAFT). However, even when all methods retain the correct label (e.g., "English Springer"), Figure 2b shows that CRAFT and ICE incur high KL divergence from the original predictions. This indicates that although the top-1 prediction is preserved, the full output distribution can still shift significantly.

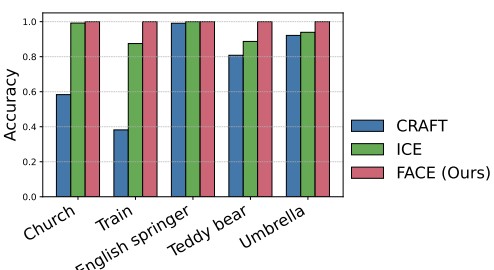 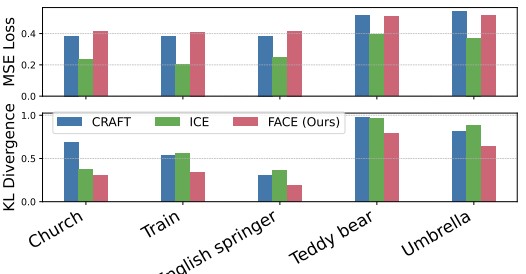

(a) Prediction accuracy on reconstructed activations. FACE consistently preserves classifier decisions across all classes, while CRAFT and ICE degrade in certain cases.

(b) MSE and KL divergence between original and reconstructed activations. Minimizing MSE alone does not ensure prediction faithfulness. High accuracy also does not imply low KL divergence, as seen in CRAFT/ICE on "English Springer."

Figure 2: Comparison of predictive alignment across concept extraction methods using ResNet-34 [14]. FACE achieves both accurate and faithful reconstructions, while CRAFT and ICE may preserve top-1 predictions yet diverge in KL-divergence.

In Figure 2b, we also note that FACE occasionally yields marginally higher MSE compared to CRAFT and ICE-an expected outcome, since our method balances the reconstruction loss with KL-regularization to preserve predictive behavior. This highlights a key tradeoff: *minimizing reconstruction loss alone is insufficient to ensure prediction faithfulness*. Unconstrained NMF may recover latent factors that resemble semantically meaningful patterns but fail to reflect class-discriminative information. By contrast, FACE explicitly enforces predictive consistency through the

KL-divergence term during factorization, ensuring that reconstructed activations preserve not only the correct decision but also the model's class confidences. This results in concept decompositions that are both interpretable and faithful (discussed in Section 4.2). We observe similar trends when evaluating on MobileNetV2 [28], reported in Appendix F.1.

## 4    Evaluation

We evaluate FACE across three key dimensions: (1) the quality of the matrix factorization, (2) the faithfulness of concept-based explanations, and (3) the complexity of the resulting explanations. We compare FACE against two state-of-the-art NMF-based concept discovery methods: ICE [36] and CRAFT [8]. We discuss evaluation with clustering based method, ACE [12] in Appendix J.

**Datasets and Models.**    We evaluate FACE on three datasets of varying semantic granularity: ImageNet [7], COCO [19], and CelebA [20]. We use ResNet-34 [14] and MobileNetV2 [28] as target models for explanation. For ImageNet and COCO, we use publicly available pretrained models. For CelebA, we train ResNet-34 and MobileNetV2 models on a curated subset of mutually exclusive binary attributes: *Black Hair*, *Blond Hair*, *Gray Hair*, and *Wearing Hat* (details in Appendix K).

**Experimental Setup.**    All results are averaged over correctly-classified 10,000 samples from 10 different ImageNet classes, 5,000 samples from 5 COCO classes, and 4,000 samples from the 4 selected CelebA attributes. Following prior work [8], we set the rank of the matrix decomposition to 25 concepts for all evaluations. We provide implementation details in Appendix B.

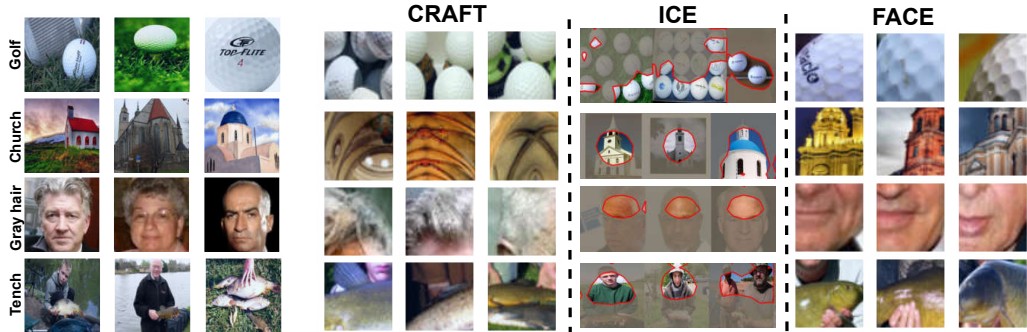

Figure 3: Qualitative comparison of top-concept extraction by CRAFT [8], ICE [36], and our method, **FACE** for four classes (Golf, Church, Gray hair, and Tench) using ResNet-34 [14].

**Qualitative evaluation.**    We provide sample explanations in Figure 3 where FACE consistently produces semantically meaningful patches. However, since FACE prioritizes faithfulness to the model's decision process, the extracted concepts may *differ* from human intuition. For instance, in the Gray hair class, CRAFT highlights hair, and ICE focuses on the forehead, but FACE reveals that the model actually relies on facial features. This highlights the importance of aligning explanations with model behavior rather than visual plausibility alone. After all, models are not constrained to use human-understandable cues; they only use features that minimize loss. See Appendix N for additional examples.

### 4.1    Quality of matrix factorization

We assess the quality of concept-based factorization along two axes: (1) reconstruction error between original $\mathbf{A}$ and reconstructed activations $\hat{\mathbf{A}} = \mathbf{U}\mathbf{W}^{\top}$ (*MSE*), and (2) prediction consistency measured as the KL divergence between model predictions on $\mathbf{A}$ vs. $\hat{\mathbf{A}}$ ($D_{KL}$). See Appendix L.1 for detailed definitions.

As shown in Table 1, FACE consistently achieves the lowest KL divergence among existing methods on minimizing the KL-divergence ($D_{KL}$) across all datasets and architectures, indicating strong alignment between the model's predictions on original and reconstructed activations. This predictive

Table 1: Quality of matrix factorization using reconstruction error (*MSE*), and prediction consistency ($D_{KL}$). Values are mean ±std across 5 runs. ↑ indicates higher is better.

| | | ResNet-34 | | MobileNetV2 | |
| --- | --- | --- | --- | --- | --- |
| | | MSE ↓ | $D_{\text{KL}}$ ↓ | MSE ↓ | $D_{\text{KL}}$ ↓ |
| **ImageNet** | ICE | **0.296 ± 0.001** | 0.359 ± 0.004 | **0.117 ± 0.000** | 0.469 ± 0.006 |
| | CRAFT | 0.451 ± 0.004 | 0.240 ± 0.003 | 0.191 ± 0.001 | 0.400 ± 0.002 |
| | FACE | 0.497 ± 0.002 | **0.220 ± 0.000** | 0.180 ± 0.001 | **0.221 ± 0.001** |
| **COCO** | ICE | **0.308 ± 0.000** | 0.596 ± 0.002 | **0.111 ± 0.000** | 0.631 ± 0.007 |
| | CRAFT | 0.457 ± 0.002 | 0.600 ± 0.004 | 0.192 ± 0.000 | 0.793 ± 0.001 |
| | FACE | 0.462 ± 0.003 | **0.458 ± 0.001** | 0.166 ± 0.001 | **0.296 ± 0.000** |
| **CelebA** | ICE | **0.148 ± 0.000** | 0.212 ± 0.000 | **0.135 ± 0.000** | 0.121 ± 0.000 |
| | CRAFT | 0.498 ± 0.010 | 0.110 ± 0.066 | 0.247 ± 0.004 | 0.160 ± 0.097 |
| | FACE | 0.375 ± 0.002 | **0.021 ± 0.001** | 0.156 ± 0.000 | **0.022 ± 0.001** |

consistency reflects the faithfulness objective encoded in our KL-regularized optimization. While FACE underperforms in reconstruction error *(MSE)*, this trade-off is expected since we minimize both MSE and KL-divergence loss. But, as shown in Section 3.4, low reconstruction error alone does not imply predictive fidelity. By explicitly incorporating prediction alignment, FACE prioritizes faithful reconstruction over raw activation proximity.

## 4.2 Faithfulness and Complexity

Table 2: Comparing faithfulness using *Concept Insertion* (*C-Ins*) and *Concept Deletion* (*C-Del*), and complexity using Gini-index sparsity (*C-Gini*). Values are mean ± std across 5 runs. ↑ indicates higher is better.

| | | ResNet-34 | | | MobileNetV2 | | |
| --- | --- | --- | --- | --- | --- | --- | --- |
| | | C-Ins ↑ | C-Del ↑ | C-Gini ↑ | C-Ins ↑ | C-Del ↑ | C-Gini ↑ |
| **ImageNet** | ICE | 0.908 ± 0.034 | 0.484 ± 0.063 | 0.537 ± 0.071 | 0.916 ± 0.020 | 0.346 ± 0.049 | 0.605 ± 0.149 |
| | CRAFT | 0.932 ± 0.001 | 0.752 ± 0.031 | 0.835 ± 0.031 | 0.886 ± 0.001 | 0.646 ± 0.024 | 0.805 ± 0.041 |
| | FACE (Ours) | **0.969 ± 0.010** | **0.891 ± 0.011** | **0.895 ± 0.001** | **0.974 ± 0.003** | **0.882 ± 0.012** | **0.947 ± 0.001** |
| **COCO** | ICE | 0.883 ± 0.029 | 0.632 ± 0.020 | 0.623 ± 0.086 | 0.906 ± 0.007 | 0.485 ± 0.051 | 0.622 ± 0.064 |
| | CRAFT | 0.861 ± 0.029 | 0.691 ± 0.029 | 0.874 ± 0.035 | 0.764 ± 0.036 | 0.571 ± 0.026 | 0.874 ± 0.047 |
| | FACE (Ours) | **0.971 ± 0.013** | **0.894 ± 0.010** | **0.947 ± 0.000** | **0.974 ± 0.002** | **0.905 ± 0.012** | **0.949 ± 0.000** |
| **CelebA** | ICE | 0.910 ± 0.008 | 0.365 ± 0.016 | 0.662 ± 0.087 | 0.858 ± 0.007 | 0.385 ± 0.050 | 0.728 ± 0.032 |
| | CRAFT | 0.953 ± 0.067 | 0.604 ± 0.036 | 0.901 ± 0.026 | 0.960 ± 0.116 | 0.592 ± 0.070 | 0.911 ± 0.028 |
| | FACE (Ours) | **0.971 ± 0.012** | **0.635 ± 0.014** | **0.928 ± 0.000** | **0.978 ± 0.001** | **0.649 ± 0.011** | **0.932 ± 0.001** |

We evaluate faithfulness using *Concept Insertion* (*C-Ins*) and *Concept Deletion* (*C-Del*), and complexity using Gini-index sparsity (*C-Gini*) as discussed in Section 3.2. As shown in Table 2, FACE consistently achieves the highest scores across all models and datasets. In particular, FACE significantly outperforms ICE and CRAFT in *C-Del*, indicating strong sensitivity of model performance to the removal of discovered concepts. *C-Ins* improvements further demonstrate that FACE's concept representations align closely with model-relevant features, enabling accurate prediction recovery when only a few key concepts are inserted. Notably, while FACE shows large margins in both *C-Ins* and *C-Del* on ImageNet and COCO, the gap is narrower on CelebA, especially for *C-Del*. We attribute this to the lower class complexity in CelebA (4 classes), where model decisions may rely on fewer features. In such cases, inserting few key concepts is often sufficient for recovering accuracy (high *C-Ins*), but removal may not fully disrupt predictions (lower *C-Del*). Lastly, the consistently high *C-Gini* scores suggest that FACE discovers compact, sparse explanations without sacrificing fidelity.

## 4.3 Ablation on regularization strength

We study the impact of KL regularization strength $\lambda$ in FACE by sweeping $\lambda$ over a wide range from $10^{-15}$ to $10^{15}$ on ImageNet [7], COCO [19], and CelebA [20] datasets using ResNet-34 [14], tracking both classifier accuracy on reconstructed activations and faithfulness with *Concept Insertion* (*C-Ins*) and *Concept Deletion* (*C-Del*).

Figure 4 shows a consistent trend across ImageNet and COCO: introducing a small KL penalty (e.g., $\lambda = 10^{-5}$) significantly improves faithfulness, as reflected by increased *C-Del* and *C-Ins*. However, for both datasets, increasing $\lambda$ beyond a moderate threshold (e.g., $\lambda \geq 10^3$) results in a sharp drop

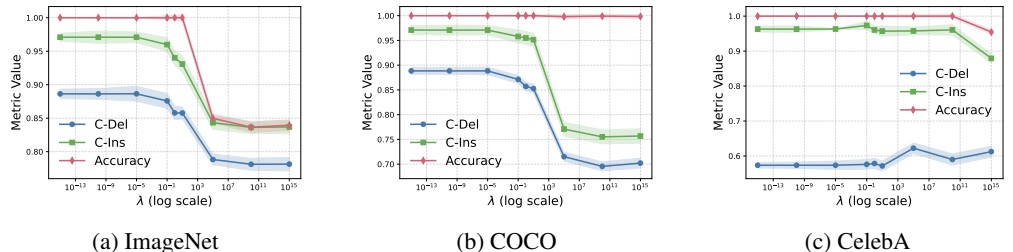

(a) ImageNet        (b) COCO        (c) CelebA

Figure 4: Effect of KL regularization strength $\lambda$ on faithfulness (*Concept Insertion* (*C-Ins*) & *Concept Deletion* (*C-Del*)) and classifier accuracy on reconstructed activations across datasets. A small KL penalty improves faithfulness across all datasets. However, large $\lambda$ values degrade performance on high-class datasets (ImageNet, COCO), while CelebA benefits from stronger regularization due to lower class complexity.

in faithfulness and, in the case of ImageNet, also harms classification accuracy. This suggests that excessive KL regularization may over-prioritize prediction alignment while compromising reconstruction fidelity.

Interestingly, we observe a different behavior on CelebA. As shown in Figure 4(c), faithfulness continues to improve up to much higher $\lambda$ values (around $10^5$), with no adverse effect on accuracy. We hypothesize that this is due to the much smaller number of classes in CelebA (4 classes), which makes it easier to optimize KL divergence between the model's output original and reconstructed activations. In contrast, ImageNet and COCO have 1000 and 200 classes respectively, which means aligning entire probability distributions is more difficult. With large $\lambda$, the optimization struggles to minimize KL effectively across high-dimensional distributions, leading to worse trade-offs. We also observe a trend of lower *C-Del* values on CelebA, which suggests that concept removal in CelebA does not change the model confidence dramatically, likely due to lower class cardinality.

These findings highlight the importance of dataset-specific calibration of $\lambda$. We present per-class trends in Appendix G and results using MobileNetV2 [28] in Appendix F.2.

## 4.4 Ablation on Matrix Decomposition Rank

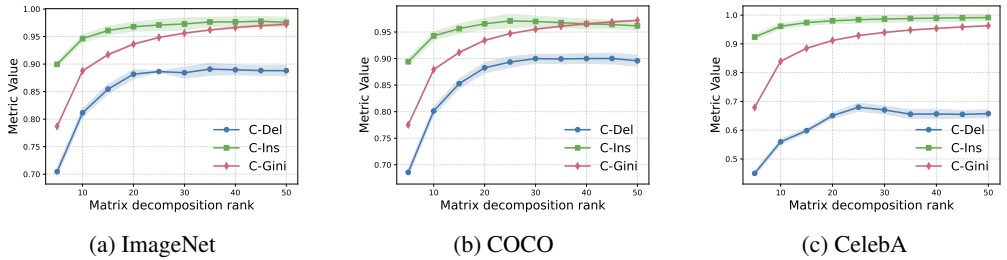

(a) ImageNet        (b) COCO        (c) CelebA

Figure 5: Effect of decomposition rank ($r$) on faithfulness (*Concept Insertion (C-Ins)*, *Concept Deletion (C-Del)* and sparsity *(C-Gini)* across datasets. Increasing rank improves both faithfulness and sparsity, with diminishing gains beyond $r = 25$.

We analyze the effect of the NMF decomposition rank ($r$), the number of learned concept dimensions, on faithfulness (*C-Ins* & *C-Del*) and sparsity *(C-Gini)*. We evaluate a range of values $r \in \{5, 10, \ldots, 50\}$ using ResNet-34 [14], with KL regularization coefficient $\lambda = 10^{-5}$ for ImageNet and COCO and $\lambda = 10^5$ for CelebA , where reconstructed activation accuracy remains 100% with high *C-Del* and *C-Ins* scores (see Figure 4). As shown in Figure 5, we observe that increasing the rank improves faithfulness metrics (*C-Ins, C-Del*) and sparsity (*C-Gini*) across all datasets. For both ImageNet and COCO, insertion and deletion scores improve sharply between $k = 5$ and $k = 25$, after which gains plateau. Meanwhile, sparsity improves with higher $k$, indicating that with more concept dimensions, activations become more selective and structured. On CelebA, we observe a similar trend, albeit with lower absolute *C-Del* values. This suggests that concept removal in CelebA

may impact model confidence less dramatically in CelebA. This is likely due to the simpler decision boundaries and lower class cardinality (4 total classes).

In practice, we use $r = 25$ similar to prior works [8]. We observe similar trends when evaluating with MobileNetV2 [28], detailed in Appendix F.3. We also present per-class trends in Appendix H. We discuss experiments on crop-size and alternative loss functions in Appendix I.2 and I.3.

**Computational cost.** FACE's optimization is lightweight, dominated by a single small matrix product $\mathbf{UW}^\top$ and a linear head, making it tractable even on low-resource hardware. Detailed runtime/memory measurements are provided in Appendix D.

## 5 Limitations

FACE is currently class-specific and global in nature, i.e., it extracts concepts for an entire class rather than on a per-instance basis. While this approach aligns with current literature [8, 9, 12, 36], it may affect the granularity of explanations in settings where instance-level interpretability is desired. Our method also relies on fixed hyperparameters (e.g., regularization strength) that require tuning for new datasets. Another limitation is the lack of a human-centered evaluation of interpretability. Since the addition of the KL loss changes the representation objective, its effect on human interpretability of the learned concepts remains unexplored. We leave such user studies, which are critical to verifying whether FACE indeed improves human understanding of latent spaces, as important future work. FACE is also suitable only to CNN based architecture. Directly applying FACE "as it is" to transformers like ViT is non-trivial. However, we see this as promising future work.

## 6 Conclusion

In this work, we presented **FACE** (Faithful Automatic Concept Extraction), a novel framework for concept-based explanations that augments Non-negative matrix factorization (NMF) with a KL divergence constraint to explicitly align concept representations with the model's predictive behavior. Unlike existing unsupervised methods that focus solely on reconstructing latent activations, FACE enforces consistency between model predictions on original and reconstructed activations, thereby ensuring the faithfulness of discovered concepts. Through theoretical analysis, we demonstrate that KL regularization bounds predictive deviation and promotes faithful local linearity in the concept space. Our empirical results across three diverse datasets (ImageNet, COCO, and CelebA) show that FACE consistently outperforms existing methods on faithfulness and sparsity.

## Acknowledgement

We thank the anonymous reviewers for their valuable comments. This research was supported by Toyota InfoTech Labs through Unrestricted Research Funds. We also acknowledge the authors of CRAFT [8] for releasing their codebase, which served as a useful foundation for this work.

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

# NeurIPS Paper Checklist

1. **Claims**

   Question: Do the main claims made in the abstract and introduction accurately reflect the paper's contributions and scope?

   Answer: [Yes]

   Justification: The abstract and introduction clearly articulate the core contribution of the paper: a novel concept-based explanation method, FACE, which augments Non-negative Matrix Factorization (NMF) with a KL divergence constraint to ensure predictive faithfulness. These claims are supported by theoretical guarantees, empirical evaluation on multiple datasets, and comparative analysis with existing methods.

   Guidelines:

   - The answer NA means that the abstract and introduction do not include the claims made in the paper.
   - The abstract and/or introduction should clearly state the claims made, including the contributions made in the paper and important assumptions and limitations. A No or NA answer to this question will not be perceived well by the reviewers.
   - The claims made should match theoretical and experimental results, and reflect how much the results can be expected to generalize to other settings.
   - It is fine to include aspirational goals as motivation as long as it is clear that these goals are not attained by the paper.

2. **Limitations**

   Question: Does the paper discuss the limitations of the work performed by the authors?

   Answer: [Yes]

   Justification: The paper includes a dedicated Limitations section (Section 5) that clearly discusses the limitations of the proposed method. It highlights the proposed method's reliance on class-level (rather than instance-level) explanations as a limitation for applications requiring fine-grained interpretability. Furthermore, the paper acknowledges sensitivity to hyperparameters like regularization strength.

   Guidelines:

   - The answer NA means that the paper has no limitation while the answer No means that the paper has limitations, but those are not discussed in the paper.
   - The authors are encouraged to create a separate "Limitations" section in their paper.
   - The paper should point out any strong assumptions and how robust the results are to violations of these assumptions (e.g., independence assumptions, noiseless settings, model well-specification, asymptotic approximations only holding locally). The authors should reflect on how these assumptions might be violated in practice and what the implications would be.
   - The authors should reflect on the scope of the claims made, e.g., if the approach was only tested on a few datasets or with a few runs. In general, empirical results often depend on implicit assumptions, which should be articulated.
   - The authors should reflect on the factors that influence the performance of the approach. For example, a facial recognition algorithm may perform poorly when image resolution is low or images are taken in low lighting. Or a speech-to-text system might not be used reliably to provide closed captions for online lectures because it fails to handle technical jargon.
   - The authors should discuss the computational efficiency of the proposed algorithms and how they scale with dataset size.
   - If applicable, the authors should discuss possible limitations of their approach to address problems of privacy and fairness.
   - While the authors might fear that complete honesty about limitations might be used by reviewers as grounds for rejection, a worse outcome might be that reviewers discover limitations that aren't acknowledged in the paper. The authors should use their best

judgment and recognize that individual actions in favor of transparency play an important role in developing norms that preserve the integrity of the community. Reviewers will be specifically instructed to not penalize honesty concerning limitations.

3. **Theory assumptions and proofs**

Question: For each theoretical result, does the paper provide the full set of assumptions and a complete (and correct) proof?

Answer: [Yes]

Justification: The paper provides a formal proposition stating that minimizing the KL divergence between the model's predictions on original and reconstructed activations bounds the total variation distance between the predictive distributions. The proposition is accompanied by a proof using Pinsker's inequality. The assumptions are clearly stated, including the definition of the predictive distributions and conditions under which KL divergence is minimized. A second theoretical subsection extends this analysis, showing how KL regularization promotes faithful local linearity even when the classifier head is linear but followed by a softmax. While this proof is more intuitive, it is supported by a Taylor expansion and formal reasoning.

Guidelines:

- The answer NA means that the paper does not include theoretical results.
- All the theorems, formulas, and proofs in the paper should be numbered and cross-referenced.
- All assumptions should be clearly stated or referenced in the statement of any theorems.
- The proofs can either appear in the main paper or the supplemental material, but if they appear in the supplemental material, the authors are encouraged to provide a short proof sketch to provide intuition.
- Inversely, any informal proof provided in the core of the paper should be complemented by formal proofs provided in appendix or supplemental material.
- Theorems and Lemmas that the proof relies upon should be properly referenced.

4. **Experimental result reproducibility**

Question: Does the paper fully disclose all the information needed to reproduce the main experimental results of the paper to the extent that it affects the main claims and/or conclusions of the paper (regardless of whether the code and data are provided or not)?

Answer: [Yes]

Justification: The paper provides complete implementation details for reproducing the main experimental results. This includes clear descriptions of dataset (ImageNet, COCO, and CelebA), model architectures (ResNet-34 and MobileNetV2), decomposition strategies (NMF with KL regularization), and concept extraction pipelines (e.g., patching, NNDSVD initialization, projection steps). Setups and hyperparameters for FACE (e.g., Adam optimizer, learning rate, early stopping criteria) are described in Appendix B, and additional training setup for CelebA is given in Appendix K. Code and scripts are made publicly available at `https://anonymous.4open.science/r/FACE-B053/`, with sample implementation notebook on a test-set available.

Guidelines:

- The answer NA means that the paper does not include experiments.
- If the paper includes experiments, a No answer to this question will not be perceived well by the reviewers: Making the paper reproducible is important, regardless of whether the code and data are provided or not.
- If the contribution is a dataset and/or model, the authors should describe the steps taken to make their results reproducible or verifiable.
- Depending on the contribution, reproducibility can be accomplished in various ways. For example, if the contribution is a novel architecture, describing the architecture fully might suffice, or if the contribution is a specific model and empirical evaluation, it may be necessary to either make it possible for others to replicate the model with the same dataset, or provide access to the model. In general. releasing code and data is often one good way to accomplish this, but reproducibility can also be provided via detailed

instructions for how to replicate the results, access to a hosted model (e.g., in the case of a large language model), releasing of a model checkpoint, or other means that are appropriate to the research performed.

- While NeurIPS does not require releasing code, the conference does require all submissions to provide some reasonable avenue for reproducibility, which may depend on the nature of the contribution. For example

    (a) If the contribution is primarily a new algorithm, the paper should make it clear how to reproduce that algorithm.

    (b) If the contribution is primarily a new model architecture, the paper should describe the architecture clearly and fully.

    (c) If the contribution is a new model (e.g., a large language model), then there should either be a way to access this model for reproducing the results or a way to reproduce the model (e.g., with an open-source dataset or instructions for how to construct the dataset).

    (d) We recognize that reproducibility may be tricky in some cases, in which case authors are welcome to describe the particular way they provide for reproducibility. In the case of closed-source models, it may be that access to the model is limited in some way (e.g., to registered users), but it should be possible for other researchers to have some path to reproducing or verifying the results.

5. **Open access to data and code**

Question: Does the paper provide open access to the data and code, with sufficient instructions to faithfully reproduce the main experimental results, as described in supplemental material?

Answer: [Yes]

Justification: The paper provides a public code repository ( `https://github.com/dipkamal/FACE`) that includes full implementation of our proposed FACE method, as well as scripts to reproduce the evaluation of FACE and baselines. The datasets used in our experiments (ImageNet, COCO, and CelebA) are publicly available. Additional training details, model decomposition are described in Appendix B.

Guidelines:

- The answer NA means that paper does not include experiments requiring code.
- Please see the NeurIPS code and data submission guidelines (`https://nips.cc/public/guides/CodeSubmissionPolicy`) for more details.
- While we encourage the release of code and data, we understand that this might not be possible, so "No" is an acceptable answer. Papers cannot be rejected simply for not including code, unless this is central to the contribution (e.g., for a new open-source benchmark).
- The instructions should contain the exact command and environment needed to run to reproduce the results. See the NeurIPS code and data submission guidelines (`https://nips.cc/public/guides/CodeSubmissionPolicy`) for more details.
- The authors should provide instructions on data access and preparation, including how to access the raw data, preprocessed data, intermediate data, and generated data, etc.
- The authors should provide scripts to reproduce all experimental results for the new proposed method and baselines. If only a subset of experiments are reproducible, they should state which ones are omitted from the script and why.
- At submission time, to preserve anonymity, the authors should release anonymized versions (if applicable).
- Providing as much information as possible in supplemental material (appended to the paper) is recommended, but including URLs to data and code is permitted.

6. **Experimental setting/details**

Question: Does the paper specify all the training and test details (e.g., data splits, hyperparameters, how they were chosen, type of optimizer, etc.) necessary to understand the results?

Answer: [Yes]

Justification: The paper specifies all experimental settings necessary to understand and reproduce the results. This includes detailed descriptions of dataset splits (e.g., 10 ImageNet classes, 5 COCO classes, and 4 CelebA attributes), model architectures (ResNet-34 and MobileNetV2), data preprocessing (e.g., image cropping with sliding windows), matrix decomposition setup (rank selection, NNDSVD initialization), training procedure (Adam optimizer, learning rate $5 \times 10^{-4}$, early stopping with $\epsilon = 10^{-3}$), and KL regularization sweep ($\lambda \in \{10^{-25}, \ldots, 10^{20}\}$). These are clearly described in the main paper and Appendix B. Additional model training details for CelebA are provided in Appendix K.

Guidelines:

- The answer NA means that the paper does not include experiments.
- The experimental setting should be presented in the core of the paper to a level of detail that is necessary to appreciate the results and make sense of them.
- The full details can be provided either with the code, in appendix, or as supplemental material.

7. **Experiment statistical significance**

Question: Does the paper report error bars suitably and correctly defined or other appropriate information about the statistical significance of the experiments?

Answer: [Yes]

Justification: All quantitative results reported in the paper include mean and standard deviation computed over five independent runs with different random seeds. These are shown in Tables 1 and 2, and error bars are clearly labeled using the "$\pm$" notation.

Guidelines:

- The answer NA means that the paper does not include experiments.
- The authors should answer "Yes" if the results are accompanied by error bars, confidence intervals, or statistical significance tests, at least for the experiments that support the main claims of the paper.
- The factors of variability that the error bars are capturing should be clearly stated (for example, train/test split, initialization, random drawing of some parameter, or overall run with given experimental conditions).
- The method for calculating the error bars should be explained (closed form formula, call to a library function, bootstrap, etc.)
- The assumptions made should be given (e.g., Normally distributed errors).
- It should be clear whether the error bar is the standard deviation or the standard error of the mean.
- It is OK to report 1-sigma error bars, but one should state it. The authors should preferably report a 2-sigma error bar than state that they have a 96% CI, if the hypothesis of Normality of errors is not verified.
- For asymmetric distributions, the authors should be careful not to show in tables or figures symmetric error bars that would yield results that are out of range (e.g. negative error rates).
- If error bars are reported in tables or plots, The authors should explain in the text how they were calculated and reference the corresponding figures or tables in the text.

8. **Experiments compute resources**

Question: For each experiment, does the paper provide sufficient information on the computer resources (type of compute workers, memory, time of execution) needed to reproduce the experiments?

Answer: [No]

Justification: The core computations for our experiments involve extracting activation features from trained models and running low-rank NMF with KL-regularization for concept extraction. All experiments were performed on a local workstation equipped with an NVIDIA TITAN Xp GPU (12 GB VRAM) running CUDA version 12.2. The average runtime for extracting concept representations with FACE per dataset was well under a minute. Given that FACE involves backpropagation through a relatively shallow classifier

head (i.e., linear layer), the compute requirements are modest and accessible on consumer-grade GPUs.

Guidelines:

- The answer NA means that the paper does not include experiments.
- The paper should indicate the type of compute workers CPU or GPU, internal cluster, or cloud provider, including relevant memory and storage.
- The paper should provide the amount of compute required for each of the individual experimental runs as well as estimate the total compute.
- The paper should disclose whether the full research project required more compute than the experiments reported in the paper (e.g., preliminary or failed experiments that didn't make it into the paper).

9. **Code of ethics**

Question: Does the research conducted in the paper conform, in every respect, with the NeurIPS Code of Ethics https://neurips.cc/public/EthicsGuidelines?

Answer: [Yes]

Justification: We have reviewed the NeurIPS Code of Ethics and ensured that our work adheres to its guidelines. The research respects principles of transparency, responsible model development, proper citation and use of external assets, and includes a dedicated discussion on broader societal impacts and potential risks. No human subjects or personally identifiable information were involved in our experiments.

Guidelines:

- The answer NA means that the authors have not reviewed the NeurIPS Code of Ethics.
- If the authors answer No, they should explain the special circumstances that require a deviation from the Code of Ethics.
- The authors should make sure to preserve anonymity (e.g., if there is a special consideration due to laws or regulations in their jurisdiction).

10. **Broader impacts**

Question: Does the paper discuss both potential positive societal impacts and negative societal impacts of the work performed?

Answer: [Yes]

Justification: We include a dedicated "Broader Impact" in Appendix A that discusses both positive and negative societal implications of our method FACE. On the positive side, we highlight how faithful concept-based explanations can improve model transparency, trust, and safety in high-stakes applications. On the negative side, we explicitly note that such interpretability techniques could be misused for adversarial purposes, such as crafting adversarial patches or targeted concept-level attacks.

Guidelines:

- The answer NA means that there is no societal impact of the work performed.
- If the authors answer NA or No, they should explain why their work has no societal impact or why the paper does not address societal impact.
- Examples of negative societal impacts include potential malicious or unintended uses (e.g., disinformation, generating fake profiles, surveillance), fairness considerations (e.g., deployment of technologies that could make decisions that unfairly impact specific groups), privacy considerations, and security considerations.
- The conference expects that many papers will be foundational research and not tied to particular applications, let alone deployments. However, if there is a direct path to any negative applications, the authors should point it out. For example, it is legitimate to point out that an improvement in the quality of generative models could be used to generate deepfakes for disinformation. On the other hand, it is not needed to point out that a generic algorithm for optimizing neural networks could enable people to train models that generate Deepfakes faster.

- The authors should consider possible harms that could arise when the technology is being used as intended and functioning correctly, harms that could arise when the technology is being used as intended but gives incorrect results, and harms following from (intentional or unintentional) misuse of the technology.
- If there are negative societal impacts, the authors could also discuss possible mitigation strategies (e.g., gated release of models, providing defenses in addition to attacks, mechanisms for monitoring misuse, mechanisms to monitor how a system learns from feedback over time, improving the efficiency and accessibility of ML).

11. **Safeguards**

Question: Does the paper describe safeguards that have been put in place for responsible release of data or models that have a high risk for misuse (e.g., pretrained language models, image generators, or scraped datasets)?

Answer: [NA]

Justification: Our paper does not involve the release of pretrained generative models, language models, or scraped datasets that pose a high risk of misuse. The released code and data pertain to concept-based interpretability for standard vision classifiers and do not require special safeguards.

Guidelines:

- The answer NA means that the paper poses no such risks.
- Released models that have a high risk for misuse or dual-use should be released with necessary safeguards to allow for controlled use of the model, for example by requiring that users adhere to usage guidelines or restrictions to access the model or implementing safety filters.
- Datasets that have been scraped from the Internet could pose safety risks. The authors should describe how they avoided releasing unsafe images.
- We recognize that providing effective safeguards is challenging, and many papers do not require this, but we encourage authors to take this into account and make a best faith effort.

12. **Licenses for existing assets**

Question: Are the creators or original owners of assets (e.g., code, data, models), used in the paper, properly credited and are the license and terms of use explicitly mentioned and properly respected?

Answer: [Yes]

Justification: All datasets (ImageNet [7], COCO [19], CelebA [20]) and pretrained models (ResNet-34 [14], MobileNetV2 [28]) used in this work are cited appropriately in the main text. These assets are publicly available and widely used in academic research under standard licenses (e.g., ImageNet: non-commercial research use, COCO: CC-BY 4.0, CelebA: for research purposes). For baseline methods (ICE [36], CRAFT [8]), we cite the original publications and adapt their publicly released code with proper attribution. All asset usage complies with respective licenses and terms of use.

Guidelines:

- The answer NA means that the paper does not use existing assets.
- The authors should cite the original paper that produced the code package or dataset.
- The authors should state which version of the asset is used and, if possible, include a URL.
- The name of the license (e.g., CC-BY 4.0) should be included for each asset.
- For scraped data from a particular source (e.g., website), the copyright and terms of service of that source should be provided.
- If assets are released, the license, copyright information, and terms of use in the package should be provided. For popular datasets, `paperswithcode.com/datasets` has curated licenses for some datasets. Their licensing guide can help determine the license of a dataset.
- For existing datasets that are re-packaged, both the original license and the license of the derived asset (if it has changed) should be provided.

- If this information is not available online, the authors are encouraged to reach out to the asset's creators.

13. **New assets**

    Question: Are new assets introduced in the paper well documented and is the documentation provided alongside the assets?

    Answer: [NA]

    Justification: The paper does not release new assets.

    Guidelines:

    - The answer NA means that the paper does not release new assets.
    - Researchers should communicate the details of the dataset/code/model as part of their submissions via structured templates. This includes details about training, license, limitations, etc.
    - The paper should discuss whether and how consent was obtained from people whose asset is used.
    - At submission time, remember to anonymize your assets (if applicable). You can either create an anonymized URL or include an anonymized zip file.

14. **Crowdsourcing and research with human subjects**

    Question: For crowdsourcing experiments and research with human subjects, does the paper include the full text of instructions given to participants and screenshots, if applicable, as well as details about compensation (if any)?

    Answer: [NA] .

    Justification: The paper does not involve crowdsourcing nor research with human subjects.

    Guidelines:

    - The answer NA means that the paper does not involve crowdsourcing nor research with human subjects.
    - Including this information in the supplemental material is fine, but if the main contribution of the paper involves human subjects, then as much detail as possible should be included in the main paper.
    - According to the NeurIPS Code of Ethics, workers involved in data collection, curation, or other labor should be paid at least the minimum wage in the country of the data collector.

15. **Institutional review board (IRB) approvals or equivalent for research with human subjects**

    Question: Does the paper describe potential risks incurred by study participants, whether such risks were disclosed to the subjects, and whether Institutional Review Board (IRB) approvals (or an equivalent approval/review based on the requirements of your country or institution) were obtained?

    Answer: [NA]

    Justification: The paper does not involve crowdsourcing nor research with human subjects.

    Guidelines:

    - The answer NA means that the paper does not involve crowdsourcing nor research with human subjects.
    - Depending on the country in which research is conducted, IRB approval (or equivalent) may be required for any human subjects research. If you obtained IRB approval, you should clearly state this in the paper.
    - We recognize that the procedures for this may vary significantly between institutions and locations, and we expect authors to adhere to the NeurIPS Code of Ethics and the guidelines for their institution.
    - For initial submissions, do not include any information that would break anonymity (if applicable), such as the institution conducting the review.

16. **Declaration of LLM usage**

Question: Does the paper describe the usage of LLMs if it is an important, original, or non-standard component of the core methods in this research? Note that if the LLM is used only for writing, editing, or formatting purposes and does not impact the core methodology, scientific rigorousness, or originality of the research, declaration is not required.

Answer: [NA]

Justification: The core method development in this research does not involve LLMs as any important, original, or non-standard components.

Guidelines:

- The answer NA means that the core method development in this research does not involve LLMs as any important, original, or non-standard components.
- Please refer to our LLM policy (`https://neurips.cc/Conferences/2025/LLM`) for what should or should not be described.

# Appendix

**Table of Contents**

# A   Broader impact

Concept-based explanations aim to bridge the gap between deep learning models and human under-standing by representing predictions in terms of semantically meaningful concepts. Our method, FACE, advances this goal by improving the faithfulness of concept-based explanations, ensuring that the extracted concepts not only look interpretable to humans but are actually aligned with the model's decision-making. This can empower users in domains such as medical diagnosis, scientific discovery, and policy reasoning, where interpretability is a prerequisite for trust and accountability, and domains where incorrect assumptions about what a model "looks at" can have significant consequences.

However, interpretability can also be misused: to justify flawed models, perpetuate biases, or manipulate user trust. While FACE improves alignment between explanations and model decisions, it cannot mitigate inherent biases in the model itself. Faithful explanations may faithfully reflect biased reasoning. Thus, FACE should be used as a tool for transparency and debugging, not as a substitute for fairness or accountability checks.

In addition, increasing access to faithful representations of a model's internal reasoning can also introduce new attack vectors. Since FACE accurately reflects the model's reasoning through sparse and interpretable concept activations, an adversary could exploit this to reverse-engineer model behavior and identify decision-critical concepts. This can aid in crafting targeted adversarial attacks that manipulate only the most influential concepts, making them more efficient and harder to detect.

# B   Implementation details: FACE

Similar to existing concept based explanation methods like CRAFT [8], ACE [12], and ICE [36], FACE also provides a global explanation to a set of images classified as a specific class. So, to explain a target class using a target network (e.g. ResNet-34), we first collect correctly classified images using the same network. We use minimum of 1000 correctly classified images of each of the 10 classes (Cassette, Chainsaw, Chruch, English springer, French horn, Garbage truck, Gaspump, Golf, Parachute, Tench) of ImageNet dataset [7], 5 classes (Hotdog, Teddy bear, Train, Umbrella, Zebra) of COCO dataset [19], and 4 classes (Black hair, Blond hair, Gray hair, Wearing hat) of CelebA [20].

Clustering-based concept method like ACE [12] generates multi-resolution segmentation of the set of images but this leads to artifacts generation. CRAFT [8] overcomes this issue by using image crops. We apply the same approach and generate local crops (patches) from each image using a sliding window approach with a kernel size $64$. We study the effect of varying patch size in Appendix I.2.

As explained in Section 3.1, a target classifier $f : \mathcal{X} \rightarrow \mathcal{Y}$ is first decomposed into two components: an encoder $g : \mathcal{X} \rightarrow \mathcal{G}$ and a classifier head $h : \mathcal{G} \rightarrow \mathcal{Y}$, where $\mathcal{G} \subseteq \mathbb{R}^p$ is an intermediate latent space. We consider two models for our analysis: ResNet-34 [14] and MobilenetV2 [28].

For ResNet-34, we define $g$ as all layers up to the second-last block, and $h$ as the final classifier head:

Listing 1: Model decomposition for ResNet34.

```
g = nn.Sequential(*list(model.children())[:-2])
h = lambda x: model.fc(torch.mean(x, (2, 3)))
```

For MobileNetV2, we use:

Listing 2: Model decomposition for MobileNet.

```
g = model.features
h(x) = model.classifier(adaptive_avg_pool2d(x))
```

The local crops (patches) are now passed through the encoder part of the network $g(\cdot)$ to extract activation maps. We apply NMF with KL-divergence regularization to the spatially averaged activation matrix $A \in \mathbb{R}^{n \times p}$, where $n$ is the number of crops and $p$ is the activation dimension. We decompose $A \approx UW^\top$ with $U \in \mathbb{R}^{n \times r}$ and $W \in \mathbb{R}^{p \times r}$, where $r$ is the number of concepts (rank). Both $U$ and $W$ are initialized using NNDSVD [4]. Our optimization objective is:

$$\min_{\mathbf{U} \geq 0, \, \mathbf{W} \geq 0} \frac{1}{2} \|\mathbf{A} - \mathbf{U}\mathbf{W}^\top\|_F^2 + \lambda \cdot \mathrm{KL}(h(\mathbf{A}) \| h(\mathbf{U}\mathbf{W}^\top)) \qquad (4)$$

We optimize this using Adam with a learning rate of $5 \times 10^{-4}$ and early stopping when the absolute change in total loss drops below below $\epsilon = 10^{-3}$. Non-negativity is enforced on $U$ and $W$ after each gradient update via in-place clamping. We sweep over $\lambda \in \{10^{-25}, \ldots, 10^{20}\}$ to select the best regularization value per dataset. We use matrix decomposition rank as 25 for experiments but provide ablation study on varying the decomposition rank hyperparameter in Section 4.4.

Our codebase is available publicly at `https://github.com/dipkamal/FACE/tree/main`.

## C  Optimization Analysis

Consider a standard supervised learning setup where a classifier $f : \mathcal{X} \to \mathcal{Y}$ maps inputs from an input space $\mathcal{X} \subseteq \mathbb{R}^d$ to output predictions in $\mathcal{Y} \subseteq \mathbb{R}^c$. Without loss of generality, we assume that $f$ can be decomposed into two components: an encoder $g : \mathcal{X} \to \mathcal{G}$ and a classifier head $h : \mathcal{G} \to \mathcal{Y}$, where $\mathcal{G} \subseteq \mathbb{R}^p$ is an intermediate latent space. The encoder $g$ maps inputs to the a high-dimensional latent representation, while $h$ maps the latent representations to the output logits, or class-scores. Consequently, $f(\mathbf{x}) = h(g(\mathbf{x}))$. Given a dataset of $n$ input samples $\mathbf{X} = [\mathbf{x}_1, \ldots, \mathbf{x}_n] \in \mathbb{R}^{n \times d}$, we denote their corresponding latent representations as $\mathbf{A} = g(\mathbf{X}) \in \mathbb{R}^{n \times p}$.

A standard NMF-based decomposition is framed as the following optimization problem:

$$\min_{\mathbf{U} \geq 0, \, \mathbf{W} \geq 0} \frac{1}{2} \|\mathbf{A} - \mathbf{U}\mathbf{W}^\top\|_F^2 \tag{5}$$

where the Frobenius norm penalizes the reconstruction error between the original activations and their low-rank approximation.

FACE augments the standard reconstruction loss with a KL divergence regularizer based on classifier head predictions:

$$L(\mathbf{U}, \mathbf{W}) = \frac{1}{2} \|\mathbf{A} - \mathbf{U}\mathbf{W}^\top\|_F^2 + \lambda \cdot \mathrm{KL}(h(\mathbf{A}) \| h(\mathbf{U}\mathbf{W})) \tag{6}$$

Standard NMF methods (e.g., multiplicative updates [17]) are not directly applicable due to the KL divergence term, which depends on the downstream classifier $h$. Instead, we optimize this objective using projected gradient descent with updates:

$$\mathbf{U}_{k+1} = \mathrm{Proj}_{\mathbb{R}_+}(\mathbf{U}_k - \eta \nabla_\mathbf{U} L(\mathbf{U}_k, \mathbf{W}_k)) \tag{7}$$

$$\mathbf{W}_{k+1} = \mathrm{Proj}_{\mathbb{R}_+}(\mathbf{W}_k - \eta \nabla_\mathbf{W} L(\mathbf{U}_k, \mathbf{W}_k)) \tag{8}$$

where $\mathrm{Proj}_{\mathbb{R}_+}$ denotes projection onto the non-negative orthant (via elementwise $\max(0, \cdot)$), and $\eta > 0$ is the learning rate.

**Convergence.**  Let us stack the decision variables as $\mathbf{Z} = [\mathbf{U}; \mathbf{W}]$ and denote the feasible set by $\Theta = \{(\mathbf{U}, \mathbf{W}) \mid \mathbf{U}, \mathbf{W} \geq 0\}$. We assume that:

$$\|\nabla L(\mathbf{Z}_k) - \nabla L(\mathbf{Z}_{k+1})\|_F \leq L_\nabla \|\mathbf{Z}_k - \mathbf{Z}_{k+1}\|_F \quad \forall \, \mathbf{Z}_k, \mathbf{Z}_{k+1} \in \Theta \tag{9}$$

for some constant $L_\nabla > 0$. Because the first term of Eqn. 6 is quadratic and the second term is a composition of smooth maps (affine head $h$ and soft-max), Eqn. 9 holds on every bounded subset of $\Theta$ with an explicit $L_\nabla$ (derived below).

By the standard descent lemma for $L_\nabla$-smooth functions, for any $\mathbf{Z}_k, \mathbf{Z}_{k+1} \in \Theta$

$$L(\mathbf{Z}_{k+1}) \leq L(\mathbf{Z}_k) + \langle \nabla L(\mathbf{Z}_k), \mathbf{Z}_{k+1} - \mathbf{Z}_k \rangle + \frac{L_\nabla}{2} \|\mathbf{Z}_{k+1} - \mathbf{Z}_k\|_F^2 \tag{10}$$

We have,

$$\mathbf{Z}_{k+1} = \mathrm{Proj}_{\mathbb{R}_+}\big(\mathbf{Z}_k - \eta \, \nabla L(\mathbf{Z}_k)\big) \tag{11}$$

Using the non-expansiveness of the projection, $\|\mathrm{Proj}(x) - \mathrm{Proj}(y)\| \leq \|x - y\|$, gives:

$$\|\mathbf{Z}_{k+1} - \mathbf{Z}_k\|_F \leq \eta \, \|\nabla L(\mathbf{Z}_k)\|_F \tag{12}$$

which combined with Eqn. 10 yields

$$L(\mathbf{Z}_{k+1}) \;\leq\; L(\mathbf{Z}_k) - \left(\eta - \tfrac{L_\nabla}{2}\eta^2\right)\|\nabla L(\mathbf{Z}_k)\|_F^2. \tag{13}$$

From this, we can see that the loss decreases as long as the term multiplying the gradient remains positive i.e. If $0 < \eta < \frac{2}{L_\nabla}$, the coefficient in Eqn. 13 is positive, so $L(\mathbf{Z}_k)$ is monotonically non-increasing and bounded below, hence convergent.

$$\eta - \tfrac{L_\nabla}{2}\eta^2 > 0 \tag{14}$$

which gives,

$$\eta < \frac{2}{L_\nabla} \tag{15}$$

**Initialization.**    We initialize $\mathbf{U}, \mathbf{W}$ using Non-negative Double SVD (NNDSVD) [4] to ensure non-negativity and leverage structure in $\mathbf{A}$, which empirically improves convergence speed, stability and quality of explanations (See Appendix I.1).

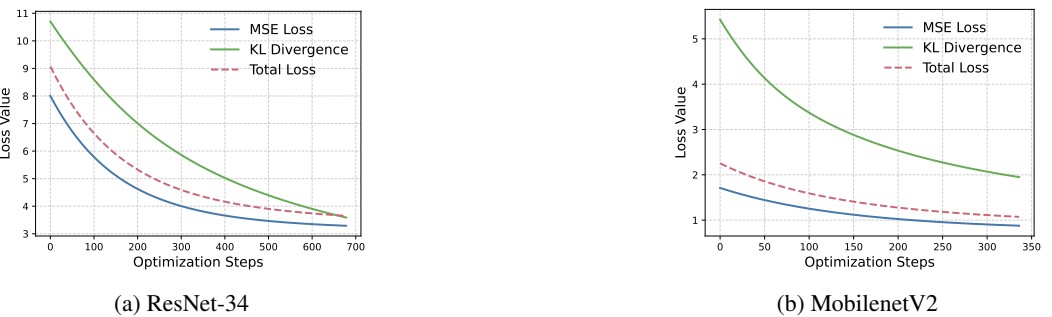

(a) ResNet-34                                   (b) MobilenetV2

Figure 6: Loss convergence during KL-regularized NMF for class 'Golf' with stopping criterion for total loss $\epsilon = 1e{-}3$

**Empirical Support.**    In practice, we observe that the loss decreases steadily and converges within few hundred or thousand steps, depending on the dataset under evaluation. See Figure 6 for a typical loss trajectory. Notably, the KL divergence continues to drop even after the MSE almost plateaus. Our optimization stops when the difference between the total losses between two consecutive steps is less than $\epsilon = 1e - 3$.

## D   Runtime and Memory Footprint

During FACE optimization, the encoder $g$ is *frozen*: gradients flow only through the low-rank factors $\mathbf{U} \in \mathbb{R}_+^{n \times r}$, $\mathbf{W} \in \mathbb{R}_+^{p \times r}$ and the (typically linear) classifier head $h$. Each PGD step forms the reconstruction $\mathbf{U}\mathbf{W}^\top \in \mathbb{R}^{n \times p}$ and applies $h$ to obtain logits.

Table 3: FACE runtime and memory on ImageNet (ResNet-34). Mean $\pm$ std across 5 runs.

| Phase | Time (s) | Peak VRAM (GB) |
|---|---|---|
| Preprocessing (activations + NNDSVD) | $5.6 \pm 0.2$ | $3.55 \pm 0.07$ |
| Optimization (PGD on $\mathbf{U}, \mathbf{W}$; $g$ frozen) | $25.7 \pm 0.2$ | $3.55 \pm 0.07$ |

We measured wall-clock time and peak VRAM on a single **NVIDIA TITAN Xp** (12 GB VRAM, CUDA 12.2) using ResNet-34, rank $r = 25$, and 1500 ImageNet images (classwise run). Preprocessing consists of computing encoder activations and NNDSVD initialization; optimization runs PGD on $(\mathbf{U}, \mathbf{W})$ with $g$ frozen. Results are provided in Table 3.

Note that FACE runtime scales with the matrix decomposition rank $r$ and the number of steps for optimization. In practice, we observed that the loss plateaus early. However, smaller decomposition rank, smaller optimization steps or early stopping approach will yield better runtime. FACE is also run *per class*; hence, the computational cost scales with the number of images analyzed for that class. For resource-constrained settings, one can subsample images per class, reduce rank $r$, or lower iteration steps without changing the method.

## E   Interpreting and controlling $\varepsilon$ in practice

By Pinsker's inequality [26], constraining the KL divergence to be less than or equal to $\varepsilon$ guarantees that the total variation distance is bounded: $\|p - q\|_1 \leq \sqrt{2 \cdot \mathrm{KL}(\cdot \| \cdot)} \leq \sqrt{2\varepsilon}$. Here, $\mathbf{p} = \mathrm{softmax}(h(\mathbf{A}))$ and $\mathbf{q} = \mathrm{softmax}(h(\mathbf{A}'))$ are the predictive distributions before and after reconstruction of activation vector. The quantity $\varepsilon$ in this inequality is *not* an assumed constant; it is the *empirical value* of $\mathrm{KL}(\mathbf{p}\|\mathbf{q})$ achieved at convergence of FACE objective given by Eqn. 16.

$$\min_{\mathbf{U} \geq 0, \mathbf{W} \geq 0} \frac{1}{2} \|\mathbf{A} - \mathbf{U}\mathbf{W}^\top\|_F^2 + \lambda \cdot \mathrm{KL}(h(\mathbf{A}) \| h(\mathbf{U}\mathbf{W}^\top)) \tag{16}$$

The hyperparameter $\lambda$ in Eqn. 16 controls the trade-off between reconstruction and prediction alignment and thereby how small $\varepsilon$ becomes. In practice we (i) monitor $\mathrm{KL}(\mathbf{p}\|\mathbf{q})$ during optimization, and (ii) select $\lambda$ from a broad "flat" region where both the KL term and the empirical $\|\mathbf{p} - \mathbf{q}\|_1$ remain small. Table 4 reports the mean KL, the mean $\ell_1$ deviation, and the Pinsker upper bound $\sqrt{2\,\mathrm{KL}}$ across a sweep of $\lambda$ (10K ImageNet images; ResNet-34; rank = 25). As discussed in ablation study of Section 4.3, large $\lambda$ hurt both the KL and reconstruction terms, whereas moderate values keep KL small and the empirical $\|\mathbf{p} - \mathbf{q}\|_1$ well within Pinsker's bound.

Table 4: **Pinsker bound in practice:** empirical prediction shift vs. $\lambda$ on 10K ImageNet images (ResNet-34). We report mean $\mathrm{KL}(\mathbf{p}\|\mathbf{q})$, mean $\|\mathbf{p} - \mathbf{q}\|_1$, and the Pinsker upper bound $\sqrt{2\,\mathrm{KL}}$.

| $\lambda$ | KL (mean) | $\|\mathbf{p} - \mathbf{q}\|_1$ (mean) | $\sqrt{2\,\mathrm{KL}}$ |
|---|---|---|---|
| $1.00 \times 10^{-25}$ | 0.302 | 0.532 | 0.777 |
| $1.00 \times 10^{-20}$ | 0.302 | 0.532 | 0.777 |
| $1.00 \times 10^{-15}$ | 0.302 | 0.532 | 0.777 |
| $1.00 \times 10^{-10}$ | 0.302 | 0.532 | 0.777 |
| $1.00 \times 10^{-05}$ | 0.302 | 0.532 | 0.777 |
| $1.00 \times 10^{-01}$ | 0.420 | 0.713 | 0.916 |
| $1.00 \times 10^{0}$ | 0.476 | 0.781 | 0.976 |
| $1.00 \times 10^{1}$ | 0.466 | 0.769 | 0.966 |
| $1.00 \times 10^{5}$ | 1.744 | 1.545 | 1.868 |
| $1.00 \times 10^{10}$ | 1.866 | 1.579 | 1.932 |
| $1.00 \times 10^{15}$ | 1.867 | 1.580 | 1.932 |
| $1.00 \times 10^{20}$ | 1.258 | 1.364 | 1.586 |

# F   Analysis on MobileNetV2

## F.1   Failure case

We investigate the limitations of standard NMF-based concept extraction methods using MobileNetV2 [28] by analyzing their predictive behavior on reconstructed activations $\hat{\mathbf{A}} = \mathbf{U}\mathbf{W}^{\top}$. As in our main results (Section 3.4), we start with a set of input images that are *correctly classified with 100% accuracy* by the original network, ensuring that any change in prediction stems solely from reconstruction artifacts.

As shown in Figure 7a, FACE consistently retains the correct predictions across all classes, while CRAFT and ICE suffer from performance degradation. For example, CRAFT achieves low accuracy on classes like "Church" and "Train." ICE performs slightly better than CRAFT but still fails to maintain full predictive consistency.

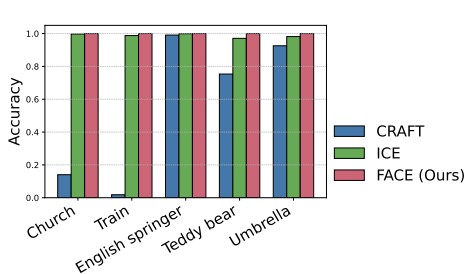

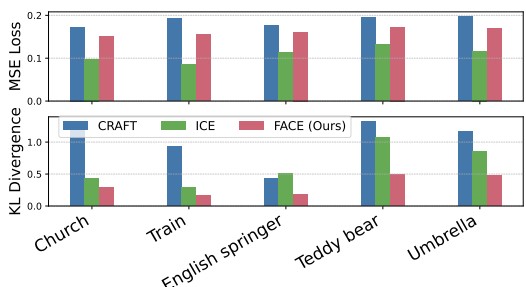

(a) Classifier accuracy on reconstructed activations using FACE, ICE, and CRAFT on MobileNetV2. FACE maintains consistent predictive behavior across all classes, while CRAFT and ICE fail.

(b) MSE and KL divergence on reconstructed activations on MobileNetV2. Minimizing MSE alone does not ensure prediction faithfulness, as seen in higher KL divergence for CRAFT and ICE.

Figure 7: Comparison of reconstruction accuracy and MSE-KL loss across concept extraction methods.

In Figure 7b, we compare the mean squared reconstruction error (MSE) and the KL divergence between predictive distributions from the original and reconstructed activations. Although CRAFT and ICE sometimes achieve lower MSE than FACE, they incur substantially higher KL divergence. This demonstrates a key limitation of unconstrained reconstruction: preserving low-level activation geometry alone does not guarantee alignment with the model's decision function. Notably, FACE achieves lower KL divergence despite slightly higher MSE, affirming that our KL-regularized optimization promotes reconstructions faithful to the model's true reasoning. These trends mirror our ResNet-34 [14] results from Section 3.4.

## F.2   Ablation: Regularization strength

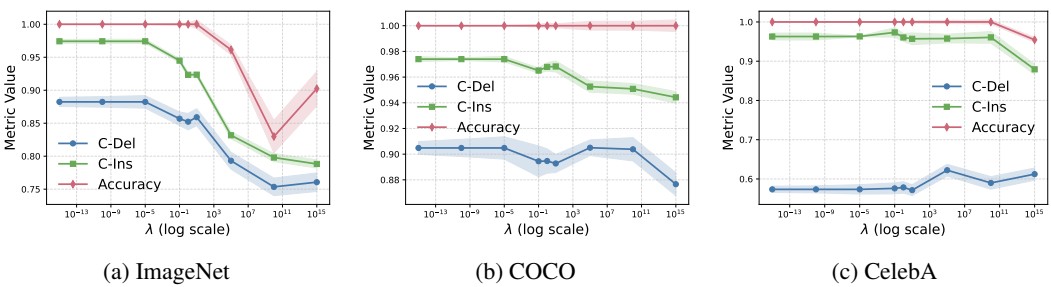

(a) ImageNet               (b) COCO               (c) CelebA

Figure 8: Effect of KL-regularization strength $\lambda$ on faithfulness (*Concept Insertion* (*C-Ins*), *Concept Deletion* (*C-Del*)) and accuracy of reconstructed activations for MobileNetV2 on ImageNet, COCO, and CelebA. Small $\lambda$ improves faithfulness, while excessively large values degrade both accuracy and explanation quality (except for CelebA, where higher $\lambda$ remains effective).

We replicate our KL regularization ablation for MobileNetV2 [28] by sweeping the regularization coefficient $\lambda$ over a range of values ($10^{-15}$ to $10^{15}$) and tracking its effect on reconstructed prediction accuracy and faithfulness metrics: *Concept Insertion* (*C-Ins*) and *Concept Deletion* (*C-Del*).

As shown in Figure 8, we observe consistent patterns across datasets. On ImageNet and COCO, introducing mild KL regularization (e.g., $\lambda \in [10^{-5}, 10^1]$) significantly boosts faithfulness, with *C-Del* and *C-Ins* improving sharply. However, when $\lambda$ becomes too large, both *C-Ins* and *C-Del* degrade, with impact on accuracy for ImageNet. This indicates that excessive KL alignment can distort the latent structure and hurt reconstruction quality.

In contrast, for CelebA, the model remains robust to higher values of $\lambda$, with faithfulness metrics continuing to improve up to $\lambda = 10^5$ without any noticeable impact on accuracy. This mirrors the results in Section 4.3. This behavior likely stems from CelebA's lower output dimensionality (4 classes), which simplifies KL optimization. In comparison, ImageNet (1000 classes) and COCO (200 classes) require aligning much higher-dimensional distributions, making it harder to simultaneously minimize KL and reconstruction losses at high $\lambda$ values.

These findings reinforce our conclusion from Section 4.3 that KL regularization plays a critical role in shaping faithful explanations, but its optimal strength depends on both the model architecture and label complexity of the dataset.

### F.3   Ablation: Rank of matrix decomposition

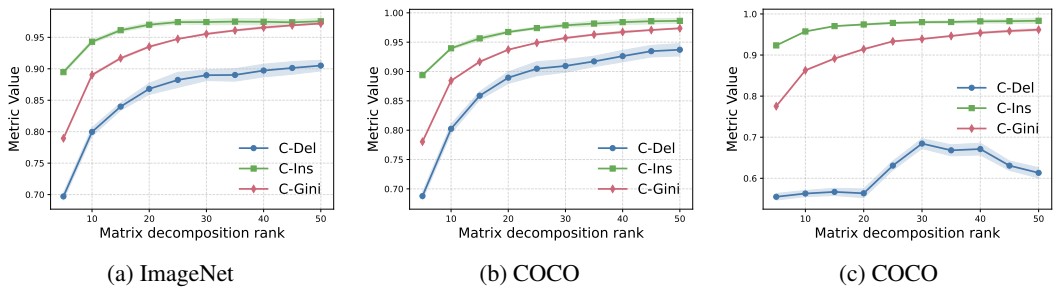

| (a) ImageNet | (b) COCO | (c) COCO |

Figure 9: Effect of NMF decomposition rank $r$ on faithfulness (*Concept Insertion* (*C-Ins*), *Concept Deletion* (*C-Del*)) and sparsity (*C-Gini*) for MobileNetV2 on ImageNet, COCO, and CelebA. Faithfulness and sparsity improve with increasing rank, stabilizing after $r = 25$.

We analyze the impact of the matrix decomposition rank $r$, the number of learned concept dimensions, on explanation quality using MobileNetV2 [28] across ImageNet, COCO, and CelebA datasets. We evaluate a range of values $r \in 5, 10, \ldots, 50$, fixing the KL regularization strength to $\lambda = 10^{-5}$ for ImageNet and COCO, and $\lambda = 10^5$ for CelebA, as these values yield 100% accuracy on reconstructed activations. and high *C-Ins* and *C-Del* score (See Figure 8).

As shown in Figure 9, we observe a consistent trend across all three datasets. Faithfulness metrics: *Concept Insertion* (*C-Ins*) and *Concept Deletion* (*C-Del*) improve significantly as rank increases from 5 to 25, after which the gains plateau. Sparsity (*C-Gini*) also improves steadily with increasing $r$. Notably, CelebA shows lower absolute *C-Del* scores than ImageNet and COCO, consistent with our observations in Section 4.4. Nonetheless, the monotonic improvement across metrics confirms that higher-rank decompositions help better capture the model's decision structure, provided the KL alignment is properly regularized.

# G   Per class-trend: Regularization strength

## G.1   ImageNet

In Figure 4a we saw that moderate KL regularization (e.g., $\lambda \in 10^{-5}$]) consistently improves faithfulness scores of concept-based explanations. Accuracy remained largely unaffected in this regime, confirming that regularization does not compromise predictive performance when appropriately tuned. However, as $\lambda$ increases beyond $10^2$, we saw a sharp drop in all three metrics, suggesting that excessive regularization distorts the feature space, leading to degraded model performance and less informative concepts.

To better understand the impact of KL regularization on individual classes, we conduct a class-wise ablation over 10 representative ImageNet classes using ResNet-34 [14], sweeping $\lambda$ across a wide range from $10^{-15}$ to $10^{15}$. For each class, we report prediction accuracy on reconstructed activations, along with *Concept Insertion* (*C-Ins*) and *Concept Deletion* (*C-Del*) as measures of explanation faithfulness.

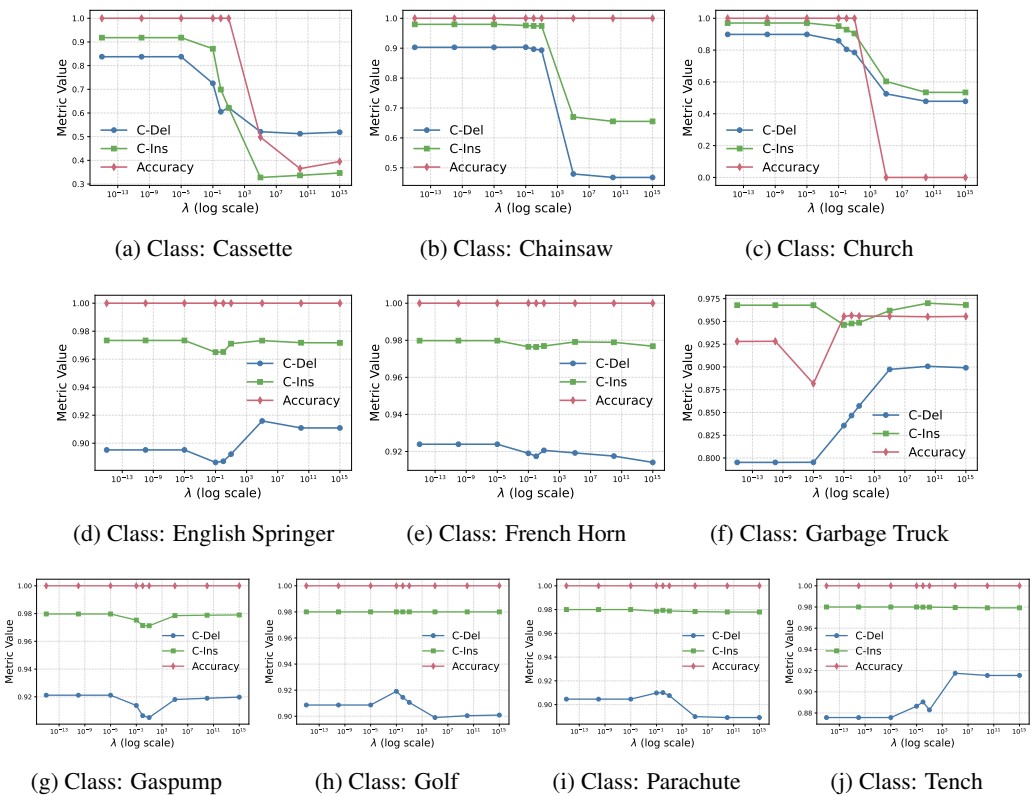

Figure 10: Class-wise ablation on KL regularization strength ($\lambda$) across 10 representative ImageNet classes. We show how *Concept Insertion* (*C-Ins*), *Concept Deletion* (*C-Del*) and accuracy vary with $\lambda$ for each class. A small value of $\lambda$ is sufficient to improve faithfulness, but large $\lambda$ often leads to degraded performance—highlighting the need for balanced regularization.

As shown in Figure 10, we observe that while the overall trend remains consistent: "low KL regularization improves faithfulness", there is notable class-dependent variability in sensitivity to $\lambda$.

Classes like Cassette, Chainsaw, and Church exhibit sharp drops in both accuracy and faithfulness for $\lambda \geq 10^3$, indicating high sensitivity to over-regularization. In contrast, classes such as English Springer, French Horn, and Parachute remain highly stable across a wide range of values. Even extremely large regularization values do not cause significant degradation. These classes may have well-separated latent structures, making it easier to align predictive behavior without sacrificing reconstruction.

These results further emphasize the need for class/dataset specific tuning of $\lambda$.

## G.2   COCO

In Figure 4b, we saw the aggregate effect of KL regularization on the COCO dataset, evaluating faithfulness and accuracy. Similar to the trend observed for ImageNet, we saw that applying a small regularization strength improves faithfulness without affecting classification accuracy.

To better understand class-specific behavior on COCO, we perform a class-wise ablation of the KL regularization coefficient $\lambda$ across five representative classes: *Hot dog*, *Teddy Bear*, *Train*, *Umbrella*, and *Zebra* using ResNet-34 [14]. As in previous sections, we sweep $\lambda$ over a wide range ($10^{-15}$ to $10^{15}$) and measure prediction accuracy on reconstructed activations along with *Concept Insertion* (*C-Ins*) and *Concept Deletion* (*C-Del*).

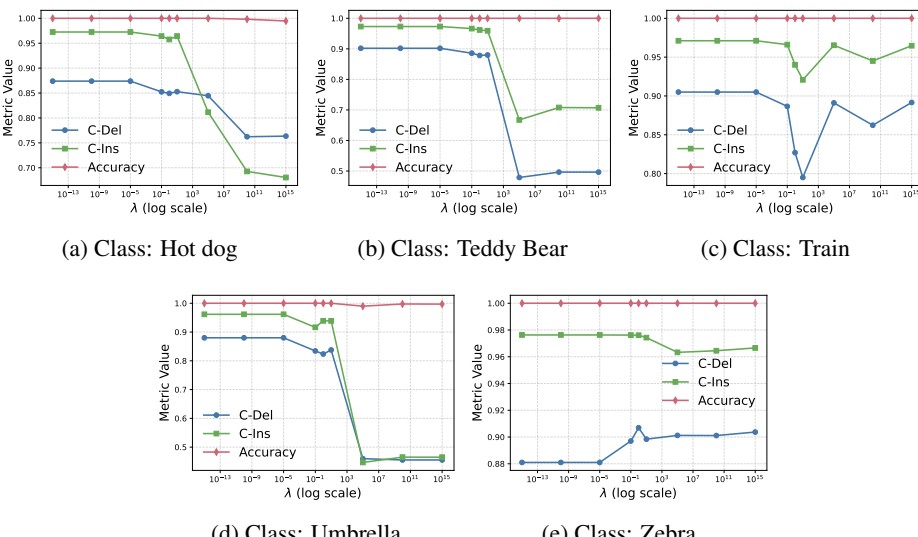

(a) Class: Hot dog          (b) Class: Teddy Bear          (c) Class: Train

(d) Class: Umbrella          (e) Class: Zebra

Figure 11: Class-wise ablation on KL regularization strength ($\lambda$) across representative COCO classes. We show how *Concept Insertion* (*C-Ins*), *Concept Deletion* (*C-Del*), and accuracy vary with $\lambda$ for each class. A small value of $\lambda$ is sufficient to improve faithfulness, but large $\lambda$ often leads to degraded performance—highlighting the need for balanced regularization.

As shown in Figure 11, we observe a consistent pattern across classes where a small KL penalty (e.g., $\lambda \in [10^{-5}, 10^{1}]$) enhances faithfulness, but excessive regularization causes sharp performance drops. *Hot dog, Teddy Bear, Umbrella* classes are particularly sensitive to large $\lambda$. *Train* class shows more robustness across a wider $\lambda$ range. Although a dip in **C-Del** occurs around $\lambda = 10^{2}$, the *Concept Insertion* (**C-Ins**) and accuracy remain mostly stable. *Zebra* class demonstrates resilience to over-regularization. Accuracy stays near-perfect across all $\lambda$, and while **C-Del** fluctuates, it does not suffer severe collapse. This maybe because of the model's reliance on clear visual features (e.g., stripes).

## G.3   CelebA

We further conduct a class-wise ablation of the KL regularization coefficient $\lambda$ across four CelebA classes, *Black hair*, *Blond hair*, *Gray hair*, and *Smiling*, using ResNet-34 [14]. For each class, we sweep $\lambda$ from $10^{-15}$ to $10^{15}$ and report classifier accuracy on reconstructed activations, along with *Concept Insertion* (*C-Ins*) and *Concept Deletion* (*C-Del*).

Unlike ImageNet and COCO, CelebA classes remain robust over a wider range of $\lambda$, with the accuracy curve staying flat up to moderately large values (e.g., $\lambda \leq 10^{10}$). Interestingly, while *C-Ins* and *C-Del* exhibit a sharp decline for large $\lambda$ in some classes (e.g., *Gray Hair*, *Wearing Hat*), other classes (e.g., *Black Hair*) show relatively stable or even slightly improved behavior until very extreme values (e.g., $\lambda \geq 10^{10}$). This suggests that CelebA's low output dimensionality (4-class prediction) makes it easier for KL minimization to enforce predictive alignment, and hence tolerate stronger regularization without substantial degradation. However, over-regularization eventually leads to underfitting in the reconstruction space, harming faithfulness.

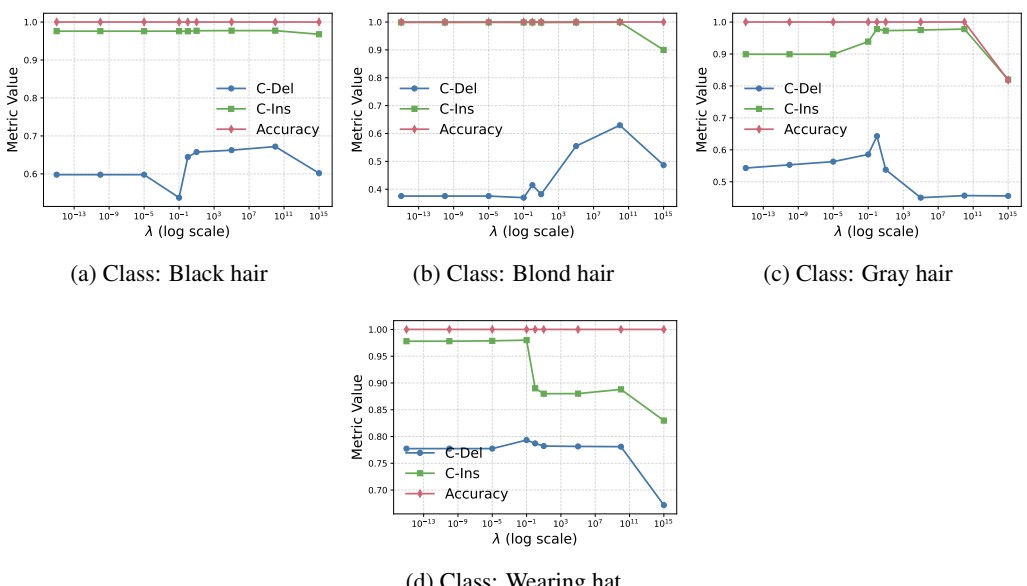

(a) Class: Black hair      (b) Class: Blond hair      (c) Class: Gray hair

(d) Class: Wearing hat

Figure 12: Class-wise ablation of KL regularization strength ($\lambda$) on CelebA on *Concept Insertion* (*C-Ins*), *Concept Deletion* (*C-Del*) and accuracy. Most classes benefit from moderate-high KL alignment, while very high regularization ($\lambda \geq 10^{12}$) causes sharp degradation in accuracy and faithfulness.

**Implications.** Our class-wise ablations across ImageNet, COCO, and CelebA reveal that the optimal KL regularization strength $\lambda$ is not universal but depends significantly on the dataset characteristics, particularly the number of output classes and the complexity of class semantics. Datasets with a large number of classes (e.g., ImageNet with 1000 classes) are more sensitive to high $\lambda$, likely due to the difficulty in aligning dense predictive distributions over many classes. In contrast, smaller datasets like CelebA-subset (4 classes) tolerate higher $\lambda$ without significant degradation in accuracy. These findings underscore the importance of dataset-specific tuning of regularization strength when aligning concept representations with model predictions.

# H    Per-class trend: Rank of matrix decomposition

## H.1    ImageNet

In Figure 5a , we observed that increasing the rank $r$ improved both *Concept Insertion (C-Ins)* and *Concept Deletion (C-Del)*, with performance saturating around $r = 25$. The class-wise analysis in the Figure 13 using ResNet-34 [14] reinforces these findings across diverse ImageNet categories where we mostly observe that *C-Ins* and *C-Del* improving sharply up to $k = 25$, after which gains plateau. In a few classes (e.g., *Garbage truck* and *Tench*) exhibits mild fluctuations beyond $r = 25$. Nonetheless, across nearly all classes, sparsity measured via *C-Gini* increases steadily with $r$. These observations support the choice of $r = 25$ as a practical balance between faithfulness and interpretability, aligning with trends in the average results.

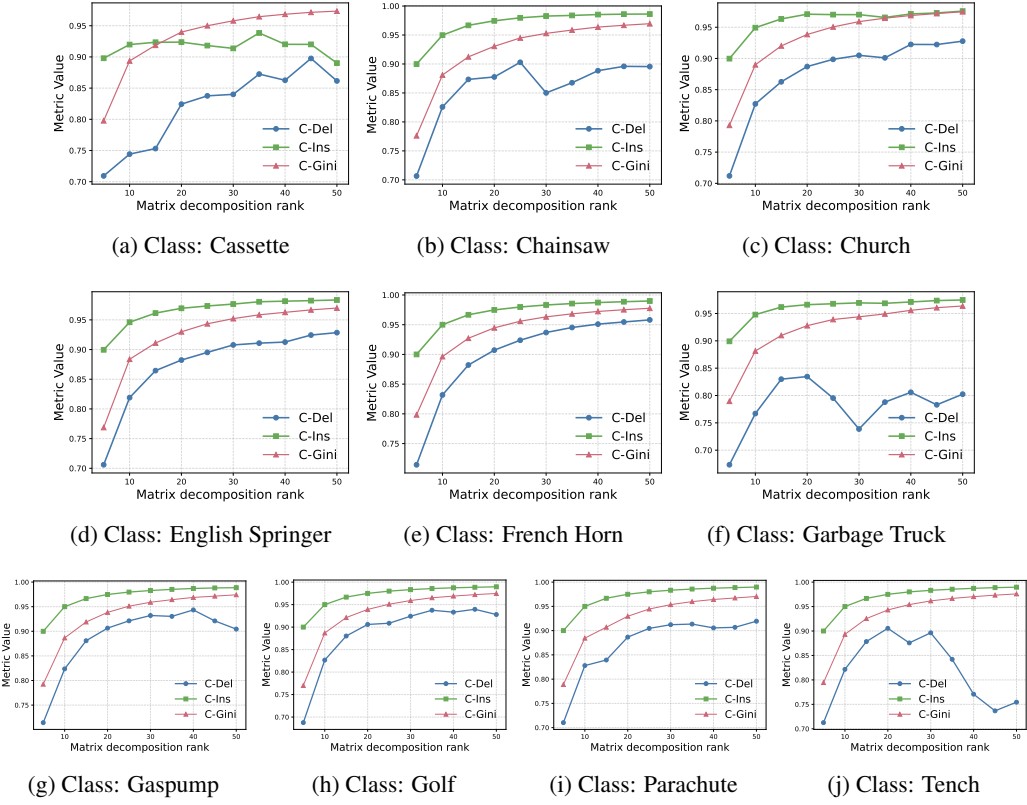

Figure 13: Class-wise ablation on rank of matrix decomposition ($r$) across representative ImageNet classes. We show how *Concept Insertion* (*C-Ins*), *Concept Deletion* (*C-Del*), and sparsity (*C-Gini*) vary with rank for each class. We observe that across most classes, faithfulness improves upto $r = 25$ and plateaus. Sparsity also improves with rank across most classes.

## H.2    COCO

In Figure 5b, we observed that increasing $r$ substantially improved both faithfulness and sparsity with improvements saturating around $r = 25$. The class-wise breakdown in Figure 14 using ResNet-34 [14] confirms this pattern across various classes of COCO dataset. Both *Concept Insertion* (*C-Ins*) and *Concept Deletion* (*C-Del*) show consistent gains in range $r = 5$ to 25, beyond which the improvements plateau. Notably, while the absolute metric values vary slightly across classes, the qualitative behavior is highly consistent, demonstrating that rank $r = 25$ provides a strong balance between faithfulness and interpretability across COCO.

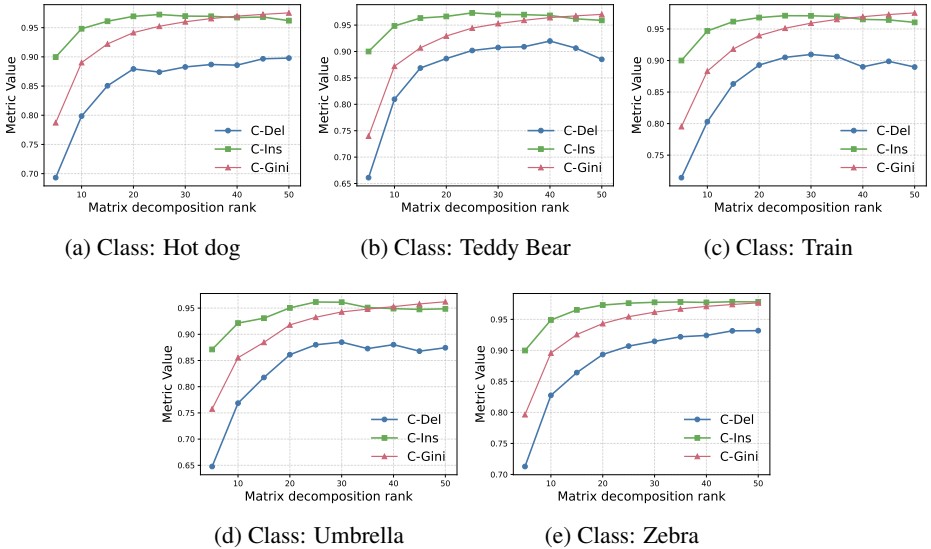

(a) Class: Hot dog  (b) Class: Teddy Bear  (c) Class: Train

(d) Class: Umbrella  (e) Class: Zebra

Figure 14: Class-wise ablation on rank of matrix decomposition ($r$) across representative COCO classes. We show how *Concept Insertion* (*C-Ins*), *Concept Deletion* (*C-Del*), and sparsity (*C-Gini*) vary with rank for each class. We observe that across most classes, faithfulness improves upto $r = 25$ and plateaus. Sparsity also improves with rank across most classes.

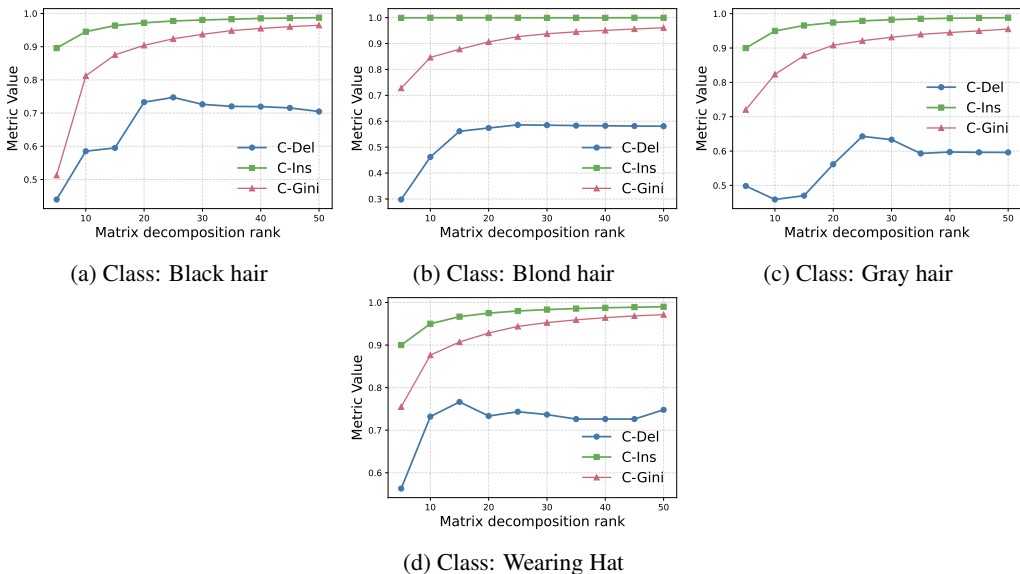

(a) Class: Black hair  (b) Class: Blond hair  (c) Class: Gray hair

(d) Class: Wearing Hat

Figure 15: Class-wise ablation on rank of matrix decomposition ($r$) across CelebA classes. We show *Concept Insertion* (*C-Ins*), *Concept Deletion* (*C-Del*), and sparsity (*C-Gini*) vary with rank for each class. We observe that across most classes, faithfulness improves upto $r = 25$ and plateaus. Sparsity also improves with rank across most classes.

## H.3 CelebA

In Figure 5c, we observed a similar trend with CelebA where the gain improved till the rank $r = 25$ although the absolute value of the *C-Del* was low. Figure 15 shows a similar trend for different classes using ResNet-34 [14]. Increasing the rank leads to steady improvements in *Concept Insertion* (*C-Ins*) and Gini-based sparsity (*C-Gini*) across most classes. However, *Concept Deletion* (*C-Del*) exhibits overall lower absolute values compared to ImageNet and COCO classes. This indicates that concept removal may impact model confidence less dramatically in CelebA, likely due to the simpler decision

boundaries and lower class cardinality (4 total classes). Still, the general trend of *C-Del* increasing and then plateauing holds. Sparsity improves consistently with rank across all classes, showing that increasing the number of concept dimensions yields more localized and disentangled representations.

# I Ablation studies

## I.1 NNDSVD initialization

To enhance the convergence and stability of FACE, we initialize the factor matrices $\mathbf{U}$ and $\mathbf{W}$ using the Non-negative Double Singular Value Decomposition (NNDSVD) algorithm [4]. NNDSVD has been shown to reduce the number of iterations required to reach a stable solution and avoid poor local minima by providing a structured low-rank approximation that respects the non-negativity constraints.

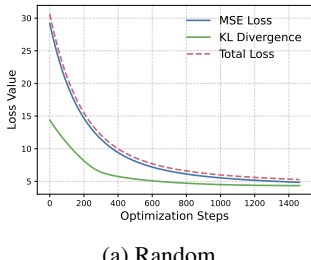
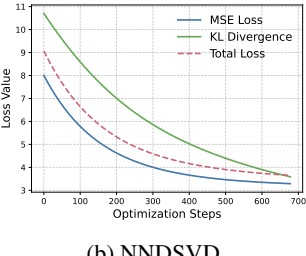

(a) Random    (b) NNDSVD

Figure 16: Loss convergence in FACE on ImageNet class "Golf" with stopping criterion for total loss $\epsilon = 1e-3$. NNDSVD initialization leads to faster convergence.

We empirically compare NNDSVD with random non-negative initialization on the ImageNet class "Golf" using ResNet-34 [14]. Figure 16 shows the optimization trajectory where we observe that compared to random initialization, NNDSVD achieves faster convergence.

Table 5: Effect of initialization method on quality of explanations for class 'Golf'. Quality of explanations is measured with **MSE** (reconstruction error between original and reconstructed activations), $D_{\mathbf{KL}}$ (KL divergence between model predictions on original and reconstructed activations), **C-Ins** (faithfulness with *Concept Insertion*), **C-Del** (faithfulness with *Concept Deletion*), **C-Gini** (sparsity of explanations) and **W-Stab** (stability of extracted concept activation vector $\mathbf{W}$ on different data subset). ↑ and ↓ indicate higher and lower are better. NNDSVD provides better reconstruction, and explanation quality.

|  | MSE ↓ | $D_{\mathbf{KL}}$ ↓ | W-Stab ↑ | C-Ins ↑ | C-Del ↑ | C-Gini ↑ |
|---|---|---|---|---|---|---|
| **Random** | $0.74 \pm 0.01$ | $1.49 \pm 0.16$ | $0.64 \pm 0.01$ | $0.92 \pm 0.03$ | $0.39 \pm 0.06$ | $0.68 \pm 0.00$ |
| **NNSVD** | $\mathbf{0.44 \pm 0.00}$ | $\mathbf{0.09 \pm 0.00}$ | $\mathbf{0.84 \pm 0.00}$ | $\mathbf{0.98 \pm 0.02}$ | $\mathbf{0.92 \pm 0.01}$ | $\mathbf{0.95 \pm 0.00}$ |

In Table 5, we summarize the final reconstruction quality, explanation faithfulness, and complexity. We can observe that compared to random initialization has significantly lower final loss, both in terms of mean squared error (MSE) and KL divergence ($D_{\mathbf{KL}}$). It also leads to better explanation stability, and higher faithfulness and sparsity results.

## I.2 Crop patch-size

We study the effect of varying the crop patch size on faithfulness and sparsity metrics for both ImageNet and COCO datasets using ResNet-34 [14], as shown in Figure 17. We observe that patch sizes larger than 60 yield consistently better performance across all evaluated metrics—including *Concept Insertion* (*C-Ins*), *Concept Deletion* (*C-Del*), and *concept sparsity* (*C-Gini*). However, increasing the patch size leads to coarser concept crops, potentially obscuring fine-grained details that are critical for interpretable explanations. To balance faithfulness with interpretability, we adopt a patch size of 64, following the design choices in prior work [8].

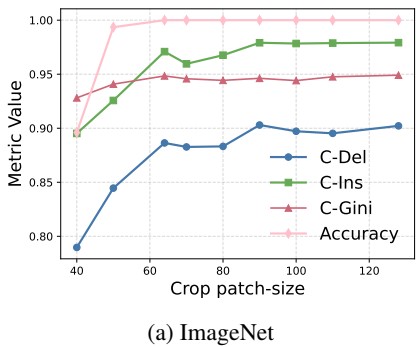

(a) ImageNet

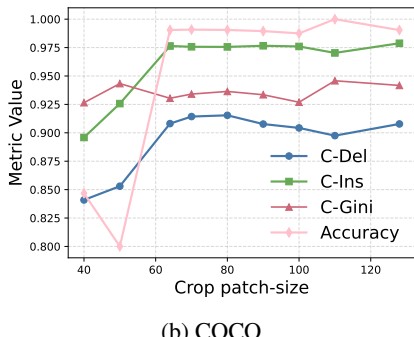

(b) COCO

Figure 17: Effect of varying crop patch size on faithfulness metrics: *Concept Insertion (C-Ins)*, *Concept Deletion (C-Del)*, Complexity with sparsity *(C-Gini)* and accuracy for ImageNet and COCO using ResNet34.

## I.3 Alternative loss functions

We chose forward KL$(p||q)$ between the original and reconstructed predictive distributions because our goal is to preserve the predictive distribution, not just the logits scale. KL-divergence operates on probabilities and gives the Pinsker bound, directly tying the objective to output deviation. As an ablation study, we swapped KL divergence loss with (i) MSE on logits and (ii) Reverse KL $(q||p)$ on 10K ImageNet images (ResNet-34).

Table 6: Ablation: Alternative loss function to KL divergence. Evaluating faithfulness of explanations using Concept Insertion (*C-Ins*) and Concept Deletion (*C-Del*). MSE represents reconstruction error of activation reconstruction and $D_{KL}$ represents prediction consistency. Values are mean across 5 runs. ↑ and ↓ indicate higher and lower are better respectively.

| Metric/Loss | Forward KL (p‖q) | Reverse KL (q‖p) | MSE Logits |
|---|---|---|---|
| **C-Ins** ↑ | **0.969** | 0.959 | 0.911 |
| **C-Del** ↑ | **0.891** | 0.875 | 0.824 |
| **MSE** ↓ | **0.497** | 0.501 | 0.525 |
| $D_{\textbf{KL}}$ ↓ | **0.220** | 0.279 | 0.315 |

Table 6 shows that forward KL $(p||q)$ gives the best faithfulness evaluation. MSE-loss preserves top-1 accuracy but allows large distribution shifts and lower faithfulness. Reverse KL sits in between. Asymmetry does not cause instability in practice because, in our optimization, p is fixed (original outputs) and q is optimized through updating $\textbf{U}$, $\textbf{W}$ matrices.

## J Comparison with ACE [12]

Our main comparison targeted NMF-based methods (ICE [36], CRAFT [8]) to isolate the effect of adding prediction alignment within the same factorization family. By contrast, ACE [12] operates in input space: it segments images into superpixels and clusters them into concepts. As noted by Fel et al. [8], superpixel segmentation can introduce boundary artifacts and fragmentation, which often dilute causal signals and negatively impacts faithfulness. While reconstruction-style scores (MSE on latent activations) are specific to factorization approaches and thus not directly applicable to ACE, we nonetheless evaluated ACE on ResNet-34 with 10K ImageNet images; the outcomes reported in Table 7 show substantially lower C-Ins/C-Del than the NMF family, with FACE strongest overall.

## K Model training details for CelebA dataset

For the purpose of evaluating concept-based explanations, we trained two classifiers on a subset of the CelebA dataset [20]. We selected a set of exclusive attribute attributes ∈ [Black Hair, Blond Hair, Gray Hair, Wearing Hat]. Each attribute was treated as a separate class with corresponding labels {0,

Table 7: Comparing faithfulness using *Concept Insertion* (*C-Ins*) and *Concept Deletion* (*C-Del*), and complexity using Gini-index sparsity (*C-Gini*). Values are mean across 5 runs. ↑ indicates higher is better respectively.

| Metrics/Method | ImageNet | | | | COCO | | | |
| --- | --- | --- | --- | --- | --- | --- | --- | --- |
| | ACE | ICE | CRAFT | FACE | ACE | ICE | CRAFT | FACE |
| **C-Ins** ↑ | 0.691 | 0.908 | 0.932 | **0.969** | 0.502 | 0.883 | 0.861 | **0.971** |
| **C-Del** ↑ | 0.510 | 0.484 | 0.752 | **0.891** | 0.577 | 0.632 | 0.691 | **0.894** |
| **C-Gini** ↑ | 0.429 | 0.537 | 0.835 | **0.895** | 0.482 | 0.623 | 0.874 | **0.947** |

1, 2, 3}. We filtered the images from the CelebA dataset based on the presence of exactly one of the selected attributes.

For model training, we used two backbone architectures pretrained on ImageNet: ResNet-34 [14] and MobileNetV2 [28]. We modified the final layers to suit our four-class classification. Both models were trained using the following settings:

- Optimizer: Adam
- Learning Rate: 1e-4
- Batch Size: 64
- Loss Function: Cross Entropy Loss
- Training Epochs: 15

Accuracy was measured on a held-out test set for each class. On test set, we obtained accuracy of 96.43% for ResNet-34 and 96.67% for MobileNetV2. Only the images correctly classified by the models were used for subsequent concept extraction and explanation evaluations.

## L   Evaluation metrics

### L.1   Evaluation of matrix factorization

1. **Reconstruction error (MSE):** We define the average reconstruction error in activation space as the Mean Squared Error (MSE) between original encoder activations $\mathbf{A}$ and reconstructed activations $\hat{\mathbf{A}} = \mathbf{U}\mathbf{W}^T$:

$$\text{MSE} = \frac{1}{np}\|\mathbf{A} - \mathbf{U}\mathbf{W}^T\|_F^2$$

where, $\mathbf{A}$ and $\hat{\mathbf{A}} \in \mathbb{R}^{n \times p}$.

2. **Prediction consistency ($D_{\mathbf{KL}}$):** We measure the prediction consistency as KL divergence loss between the original model predictions $h(\mathbf{A})$ and the model predictions on the reconstructed activation $h(\mathbf{U}\mathbf{W})$:

$$D_{\text{KL}} = \text{KL}\left(\text{softmax}(h(\mathbf{A}))\|\text{softmax}(h(\mathbf{U}\mathbf{W}^T))\right)$$

### L.2   Evaluation of concept explanations

1. **Faithfulness:** We evaluate the faithfulness of concept-based explanations using perturbation-based metrics adapted from the AOPC framework [27], widely used in feature attribution literature.

   **(a) Concept Deletion (C-Del):**   Given a set of concept importance scores $I(c_i)$ and a batch of concept representations $\mathbf{U} \in \mathbb{R}^{n \times H \times W \times r}$, we progressively set the top-$k$ most important concepts to zero in $\mathbf{U}$, compute the reconstructed activations $\tilde{\mathbf{A}} = (\mathbf{U} \odot M_k)\mathbf{W}^T$, and evaluate classification accuracy on the perturbed representations:

$$\text{Accuracy}_k = \frac{1}{n}\sum_{i=1}^{n}\mathbb{1}\left[\arg\max h(\tilde{\mathbf{A}}_i) = y_i\right]$$

where $M_k$ is a binary mask that zeroes out the top-$k$ concepts across all spatial locations. The Concept Deletion score is computed as the area over the accuracy curve:

$$\text{C-Del} = \frac{1}{r+1} \sum_{k=1}^{r} \left( \text{Accuracy}_0 - \text{Accuracy}_k \right)$$

where $r$ is the total number of concepts, and $\text{Accuracy}_0$ is the accuracy without deletion. Higher C-Del scores imply greater performance degradation upon removing important concepts, indicating higher faithfulness. If accuracy remains unchanged after removing top concepts, the score is low, suggesting the attribution is unfaithful.

**(b) Concept Insertion (C-Ins):** Complementing deletion, Concept insertion (C-Ins) measures how rapidly accuracy recovers when important concepts are added incrementally. Starting with an all-zero concept tensor $\mathbf{U}_0 = \mathbf{0}$, we progressively insert the top-$k$ most important concepts and compute the reconstructed activations:

$$\tilde{\mathbf{A}} = (\mathbf{U}_0 + M_k \odot \mathbf{U})\mathbf{W}^\top$$

The model's classification accuracy is evaluated at each step $k$. The Concept Insertion score is then defined as the area under the accuracy curve:

$$\text{C-Ins} = \frac{1}{r} \int_0^r \text{Accuracy}_k \, \mathrm{d}k$$

In practice, we approximate this integral numerically using the trapezoidal rule. Higher C-Ins scores indicate faster recovery of accuracy from fewer concepts, implying that the explanation identifies truly influential features.

2. **Complexity (C-Gini):** We use the definition of Gini-index for sparsity to measure complexity of explanations, as used by Chalasani et al. [5]. Given a classifier $f$, a concept-based explanation method $m$, and a set of extracted concepts $\{c_1, c_2, ..., c_n\}$ with corresponding importance scores $I(c_i)$, we define the complexity of $m$ as:

$$\text{C-Gini} = \frac{\sum_{i=1}^{n}(2i - n - 1) \cdot \text{sorted}(I(c_i))}{n \cdot \sum_{i=1}^{n} \text{sorted}(I(c_i))}. \tag{17}$$

Here, $I(c_i)$ represents the absolute importance score of concept $c_i$, $n$ is the total number of extracted concepts, $\text{sorted}(I(c_i))$ denotes the concept importance scores sorted in ascending order. C-Gini values lie in between $[0, 1]$. A value of 1 indicates perfect sparsity (low complexity), where only one element in the vector $\phi_i(\mathbf{x}) > 0$. The sparsity is zero if all the vectors are equal to some positive value, which means high complexity.

# M  Concept importance using Sobol-Indices

To quantify the importance of extracted concepts with respect to a model's prediction, we use Sobol indices [32] instead of TCAV [16] as they lead to better estimates of important concepts, as demonstrated by Fel et al. [8].

The Sobol sensitivity index $S_u$ measures the contribution of a specific concept subset $U_u$ to the output variability of the model $f(U)$, where $U$ denotes the full set of input concepts. In particular, $S_u$ captures how much of the variance in the model output is attributable to the subset $U_u$, thus providing an interpretable metric of concept importance in terms of functional fluctuation.

The sensitivity index $S_u$ is formally defined as:

$$S_u = \frac{\mathbb{V}(f_u(U_u))}{\mathbb{V}(f(U))} = \frac{\mathbb{V}(\mathbb{E}(f(U)|U_u)) - \sum_{v \subset u} \mathbb{V}(\mathbb{E}(f(U)|U_v))}{\mathbb{V}(f(U))},$$

where $\mathbb{V}$ denotes variance and $\mathbb{E}$ denotes expectation. This formulation decomposes the total output variance of $f(U)$ into contributions from each subset of input concepts and their interactions.

Hence, Sobol index for a concept-subset is defined as the proportion of total output variance explained by that subset. In practice, we often use the Total Sobol Index $S_i^T$ to assess the overall importance

of a concept $i$, which accounts for both its individual contribution and all its interactions with other concepts. The total index is given by:

$$S_i^T = \sum_{u \subseteq \mathcal{U}, i \in u} S_u.$$

To compute these indices, Fel et al. [8] employs the Jansen estimator [15] along with a Quasi-Monte Carlo Sequence (Sobol $LP_\tau$ sequence) to efficiently approximate the necessary variance and expectation terms. We refer the readers to the paper CRAFT [8] for more details.

# N  Qualitative results

Figure 18 compares concept extraction by CRAFT [8], ICE [36] and FACE across sample examples of ImageNet, COCO and CelebA dataset using ResNet-34.

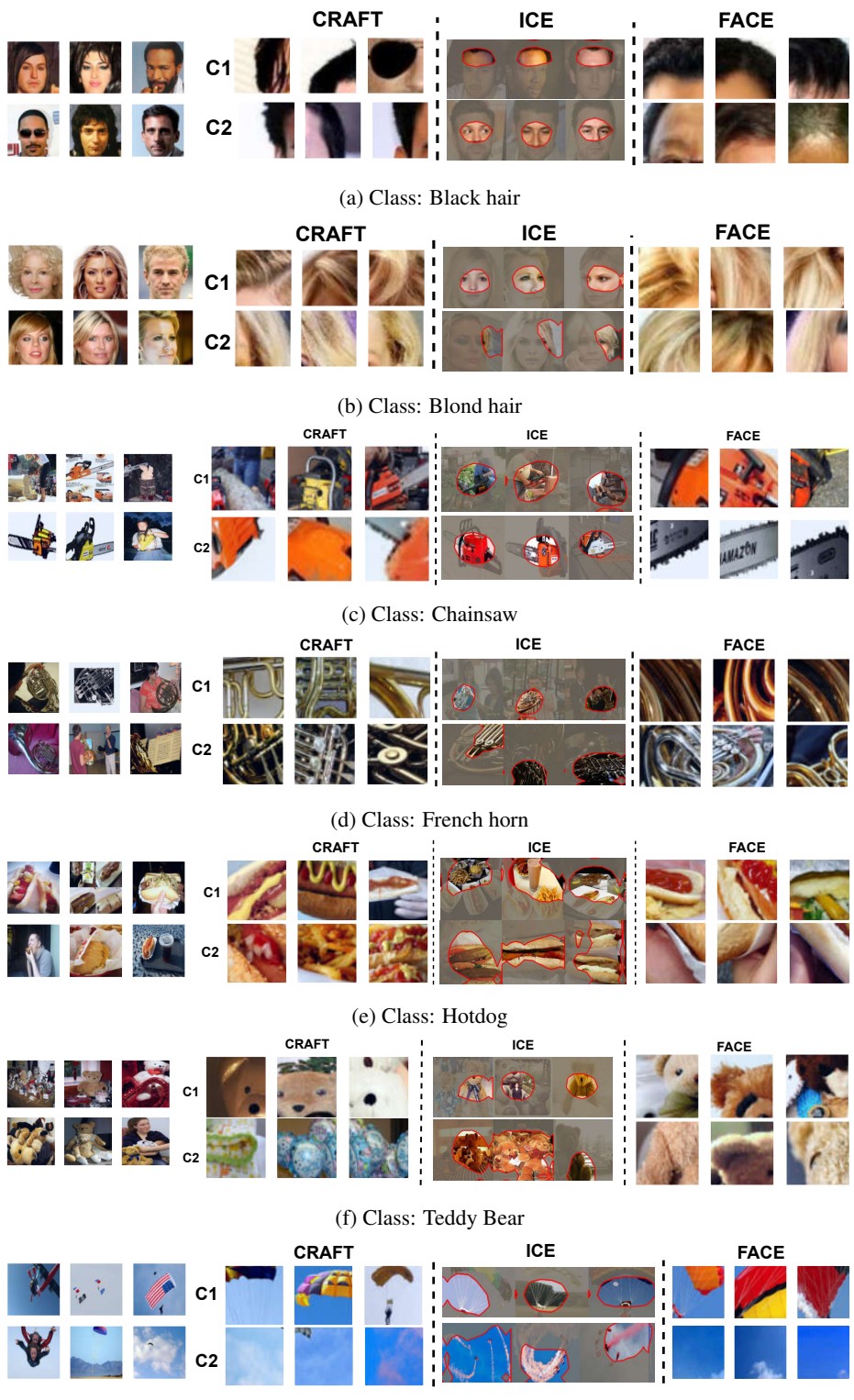

Figure 18: Concept-based visualizations for different classes.

