# OpenReview forum: "FACE: Faithful Automatic Concept Extraction"
_NeurIPS.cc/2025/Conference — NeurIPS 2025 poster_

### Official Review · Reviewer_FuWA · 2025-06-23

**Clarity:** 3
**Significance:** 4
**Originality:** 3
**Rating:** 4
**Confidence:** 4

**Summary:**

This paper aims to address an identified weak spot in concept-based XAI, which arises from neglecting the alignment of extracted concept representations with the model's inference process. The authors propose FACE (Faithful Automatic Concept Extraction), which elegantly combines the reconstruction loss of a sought NMF solution with a Kullback-Leibler divergence term to guide the optimization process toward minimizing deviations in the model's inference behavior. While the reviewer believes that the paper is a step in the right direction with an already solid foundation, its grounding in related work and the robustness of the evaluation could be improved.

**Questions:**

*Questions*
- FACE does not appear to be mathematically limited to application on the penultimate layer of the model, yet the setup may lead the reader to believe so (e.g., the use of terms like "classifier head," see line 87). Have you applied the method to other intermediate layers of the model(s), and compared the results to those of related approaches?
- The faithfulness evaluations described in Section 3.2 are performed at the granularity of model accuracy. This means you will only observe measurable effects if there are sufficient data points for which the model changes its predicted class. Did you consider measuring (and averaging) changes in output logits for each individual data point instead? This way, the imposed change does not need to be dominant enough to affect the softmax output and aggregate over a large sample to become quantifiable. This more granular and sensitive approach would better reflect the true effect of each ablation step on the model, and would be easier to observe (since it is not obscured by the softmax layer).
- Equation 2: Did you consider using alternatives to KL divergence as baselines for preserving model inference behavior—e.g., simple MSE (potentially on pre-softmax logits) or sample-wise cross-entropy—which are both symmetric, in contrast to KL divergence?
- How does the asymmetric nature of KL divergence affect the optimization of the FACE objective?
- Consider moving the discussion of Figure 2a, currently found at lines 226ff, to Section 4.1, where the relevant context is introduced.
- Typo in line 723: It should read FACE underperforms, not slightly underperforms.
- The qualitative results in Figure 3 are discussed in a very subjective manner. To the reviewer, all methods' explanations appear plausible, while FACE might have found a shortcut in the particular facial features it highlights. It would be highly interesting to further investigate the differing concept representation qualities of each method to better understand their respective utilities in uncovering meaningful aspects of model inference behavior.
- Table 1: U-Spr should include a downward-pointing arrow, as the text describes the metric measures the "average fraction of active (non-zero) concepts." A high value would indicate a dense feature encoding with low sparsity.
- Table 2: Please unify the number formatting for consistency and clarity.
- Have you considered directly comparing FACE to ACE in the main paper, given that ACE is mentioned in the appendix? Including it in the main comparison would strengthen the empirical analysis.

**Ethical Concerns:**

["NO or VERY MINOR ethics concerns only"]

**Final Justification:**

the authors have shown that the flaws listed in my review, leading to a disfavourable rating, can be fixed for a camera ready version of the paper, should it be accepted successfully. Still, AC should check whether the non-anonymized code submitted with the paper will have an impact on the final decision.

**Limitations:**

yes.

**Quality:**

3

**Strengths And Weaknesses:**

*Strengths*
- The proposed solution is simple and elegant, and can be easily combined with various other post-hoc, concept-based explanation approaches.
- The authors have delivered clean and well-structured code for reproducibility.
- The use of faithfulness testing through concept insertion and deletion as a quantitative means to estimate concept quality is a good choice.
- Overall, the paper is well written, and the setup of the experiments is logical, although not fully developed or completed.


*Weaknesses*
- The paper is focused on a narrow range of related work, revolving around direct comparisons with methods ACE, ICE, and CRAFT. This leaves the impression that all works in this area solely focus on reconstruction for concept extraction while completely ignoring the model's use of those features. Since all presented and compared methods aim to communicate the semantics of latent features via examples, the reviewer recommends extending the scope of the related work and the comparative discussion to also include Relevance Maximization (RM) from [Achtibat et al., 2023]. While RM does not attempt to identify influential concepts via reconstruction using NMFs or SAEs and instead relies on naturally emerged neurons (a limitation acknowledged by the authors), it proposes an alternative to the widely-used Activation Maximization (AM) approach. RM selects representative examples for neurons/features, primarily reflecting the model's use of them during inference. As such, RM constitutes a contrasting perspective—no reconstruction but a focus on model behavior vs. reconstruction with limited or no incorporation of model inference—on concept representation, which could optionally be combined with all compared approaches. In the end, RM follows the same goals as FACE and would thus be a valuable addition to the discussion.
- Figure 1 is missing important information. For one, how were concepts C1 and C2 selected? Furthermore, the representational misalignment evaluation is purely qualitative, and the claim that FACE highlights concepts more aligned with model reasoning than the compared approaches should be supported with quantitative data in the figure.
- The concept deletion and insertion evaluation is based on model accuracy, which renders the evaluation procedure insensitive to small changes in the model during ablation. See Questions section.
- Since the primary goal of FACE is to help end-users better understand neural network predictions, and its major effect on interpretability is via the augmentation of presented examples to communicate concepts, a user study is critically missing. This component would elevate the more qualitative aspects of the paper’s evaluation.
- Lines 178+ and 207+ use inconsistent notations for Jacobian matrices, i.e., $\nabla$ and $J$.
- It is unclear whether the 4k to 10k samples per dataset are used for the optimization of each method, whether there is a hold-out set, or whether both training and testing of the concept representation are performed on the same sample set. This information is crucial for interpreting the results in Section 4.3. Based on lines 307ff, the observed results could be explained by the KL term causing the FACE solution to overfit on the training subset—if such a subset exists.
- In the context of Section 4.2, it is not clearly described how "concept importance" is measured. Do the authors directly use activation as a measure of importance (which previous literature, e.g., [Achtibat et al., 2023], shows is not correlated with concept utility), or do they use another method?
- The results in Figure 4 / Section 4.3 are missing a proper baseline for comparison. For example, if accuracy with COCO remains largely unaffected regardless of the choice of $\lambda$, does the added KL term in the FACE optimization have any effect at all? How do ICE or CRAFT perform under the same settings?

*Remark*
- The anonymized code base is not fully anonymized, as the comments at the head of the FACE related files enable the identification of the author(s)

---

> ### Author Rebuttal · Authors · 2025-07-30
>
> We sincerely thank the reviewer for their thoughtful review and for highlighting that FACE proposes an elegant solution to addresse a weak spot in concept-based XAI. Below, we clarify some concerns:
> - **Broader context:** We agree with the reviewer that focusing only on ACE/CRAFT/ICE can make it seem as if all automatic concept methods ignore model usage. While our focus was on automatic concept extraction methods that followed similar factorization-based techniques, we will explicitly discuss Relevance Maximization in the related‑work section, highlighting the conceptual contrast to FACE. RM is “no reconstruction + direct model behavior,” while FACE is “reconstruction with explicit prediction alignment via KL.” Both aim to expose what the model actually uses, but through different techniques.
>
> - **Fig. 1 Concepts chosen and quantitative support:** In Fig. 1, concepts C1​, C2​ were simply the top Sobol‑importance concepts for each method. We will state this explicitly in the caption/text. The qualitative “misalignment” claim is supported by our faithfulness metrics (C‑Ins/C‑Del) and KL on outputs (Tables 1–2). We will add the class-specific scores in the caption.
>
> - **User study:** Our primary target is predictive faithfulness, not human plausibility. Human studies are essential when concepts are supposed to match human semantics (e.g., concept-bottleneck models). In contrast, FACE purposefully reveals potentially “non‑plausible” but model‑critical cues (shortcuts, biases). We will make this rationale explicit. A user study, while valuable for downstream interpretability interfaces, is beyond the scope of this faithfulness-focused work. We will add this to our current limitation.
>
> - **Penulimate layer limitation?** FACE is not restricted to the penultimate layer. For any intermediate layer $l$, we simply have to define $g_{\le \ell}$​ as the network up to that layer and $h_{>\ell}$ as the remaining layers. We focused on the penultimate layer to match prior works and, most importantly, to target higher-level, semantically coherent concepts because, as activation merge towards the final layer, more high-level concepts are revealed. We have verified this qualitatively as extracted concepts shift from textures/edges/patterns in early blocks to specific high-level parts (eg, body parts like ears, fur for animals). Quantitatively, faithfulness metrics are the lowest in the earlier layers, consistent with Fel et al. Neurips 2024. We can add this discussion in Appendix.
>
> - **Accuracy granularity vs. logits?** We agree that logit-level signals are more sensitive. Note, however, that our evaluation already includes a distribution-level measure: the KL divergence between original and perturbed predictions, which is computed on the full softmax output, not on accuracy. Still, to directly address the reviewer’s point, we computed additional logit-level insertion/deletion curves on 10K ImageNet images (ResNet34) and got the following scores:
> | Metrics/Method    | FACE | CRAFT | ICE |
> | -------- | ------- | ------- | ------- |
> | C-Ins  | 0.9466    | 0.8386 | 0.6403 |
> | C-Del | 0.9472     | 0.8413 | 0.6448 |
>
> We can add this additional evaluation in the Appendix for all datasets and models.
> - **Alternative to KL and asymmetry impact:** We chose forward KL(p∥q) between the original and reconstructed predictive distributions because our goal is to preserve the predictive distribution, not just the logits scale. KL operates on probabilities and gives the Pinsker bound used in §3.3, directly tying the objective to output deviation. Still, for completeness, we swapped current-KL loss with (i) MSE on logits and (ii) Reverse KL (q||p) on 10K ImageNet images (Resnet34). Forward KL (p||q) gave the best faithfulness evaluation. MSE preserved top-1 accuracy but allowed large distribution shifts and lower faithfulness. Reverse KL sat in between. Asymmetry did not cause instability in practice because, in our optimization, p is fixed (original outputs) and q is optimized through updating U, W matrices. Table below shows the result. We will add this ablation to the appendix.
>
> | Metrics/Loss    | Original forward KL (p\|\|q) | MSE logits loss | Reverse KL (q\|\|p) |
> | -------- | ------- | ------- | ------- |
> | C-Ins  | 0.969    | 0.911 | 0.959 |
> | C-Del | 0.891     | 0.824 | 0.875 |
> | MSE | 0.497    | 0.525 | 0.501 |
> | KL | 0.220     | 0.315 | 0.279 |
>
> - **On U-Spr metric:**  We thank the reviewer for catching the ambiguity. Our current definition of U-Spr does measure the active fraction (density), so lower is better for sparsity. Because our projected gradient descent updates with non‑negativity rarely drive entries exactly to zero, FACE ends up with many very small but non-zero coefficients, yielding higher density than ICE/CRAFT, whose multiplicative updates often drive many entries to zero, so they look “sparser”. Two simple fixes are possible (we can apply one in the camera-ready): a) report density explicitly as U-Den (downarrow) or b) report sparsity as 1-U_Spr. This correction does not change our final conclusion: *FACE still achieves the lowest KL (prediction consistency) and strongest C-Del/C-Ins faithfulness. Moreover, the explanation-level sparsity, captured by C-Gini in concept importance, remains the highest in FACE. Our goal is faithful, succinct explanations, not necessarily the sparsest latent code.*
> - **On ACE evaluation:**  We thank the reviewer for the suggestion. Our main comparison targeted NMF-based methods to isolate the effect of adding prediction alignment to the same factorization family. ACE relies on superpixel segmentation and uses a clustering-based approach. As noted by Fel et al. (CRAFT), this can introduce artifacts that degrade faithfulness metrics. While the reconstruction-based metrics are specific to factorization-based techniques, we evaluated ACE on ResNet34 10K ImageNet images, and got the following result:
> | Metrics/Method    | FACE | CRAFT | ICE | ACE |
> | -------- | ------- | ------- | ------- |------- |
> | C-Ins  | 0.969    | 0.0.932 | 0.908 |0.691 |
> | C-Del | 0.891     | 0.752 | 0.484 | 0.510|
> | C-Gini | 0.895 | 0.835 | 0.537 | 0.429|
>
> We can add ACE analysis on all dataset and models to the appendix.
> - **Subjectivity in Fig. 3 and possible shortcuts:**  We agree that qualitative judgments can be subjective. Our C‑Del test directly probes causality: zeroing FACE’s top “face-feature” concepts in CelebA Gray Hair sharply drops logits/accuracy, while removing low-importance concepts barely moves predictions. We ran an additional experiment to inspect if FACE captures the explicit bias in the dataset. We created adversarial patch images that force misclassification on ImageNet. FACE isolates the patch as the top concept, identifying the shortcut cues used by the model.
> - **Concept importance:** We do not use raw activation magnitudes. Instead, we follow CRAFT and estimate variance-based Sobol importance using Halton sampling. This measures how much each concept contributes to the output variance when masked/unmasked, aligned with model behavior. This is more robust than activation alone. We have mentioned this in Line 100-101 and will make this explicit wherever introduced.
> - **Data usage, splits, overfitting:**  FACE never updates the classifier; it only factorizes fixed latent activations with a low‑rank NMF. Following prior concept‑XAI work (ACE/CRAFT/ICE), we take a class of correctly classified images we want to explain for a given model (this comes from a test-set and not a training set). We then fit U,W on the set of images to explain the class of prediction (e.g., all correctly classified ImageNet “church” images). The computed low-rank matrices U and W are now used to explain the model prediction of “church”. For CelebA, we train the model using a train set and only compute concepts on a test set.
> -  **The results in Figure 4 / Section 4.3:** In the sweep (Fig. 4), once λ exceeds ~10³, both C‑Ins and C‑Del drop sharply, showing the KL term is doing work: too much weight on KL harms the factorization’s ability to isolate causal concepts. Accuracy stays high because the model can still guess the right label, as Accuracy is a step function; large logit drops can occur without flipping the argmax, so accuracy barely moves.  This is exactly the situation where your proposed logit‑based evaluation is more sensitive, and we adopted it. The table below shows the logit-based AUC evaluation:
> | Metrics/Method    | FACE | CRAFT | ICE |
> | -------- | ------- | ------- | ------- |
> | C-Ins  | 0.961    | 0.761 | 0.668 |
> | C-Del | 0.934    | 0.781 | 0.722 |
> - **We will also incorporate other suggestions you mentioned regarding notation, formatting and placement of some sections.**

---

> > ### Comment · Reviewer_FuWA · 2025-08-06
> >
> > thank you for the clarifications and carefully addressing my comments and running additional experiments.
> > As a result of the improvements made, **I would increase my score to 4** and suggest the authors to incorporate the raised points into the camera ready version of their paper, should it be accepted.
> >
> > Comment on the User Study discussion: While I do agree that the user study is out of scope for this paper. still, I firmly believe that such a study should be considered in follow-up work, as the whole motivation of using SAEs/NMFs/similar stems from the need to make the latent space humanly accessible by disentangling and transforming concept representations. since the addition of KL loss changes the objective for the representation layer, the effect on human interpretability should be monitored.

---

> ### Author Response · Authors · 2025-08-07
>
> We sincerely thank the reviewer for the positive feedback and raising your score. We will incorporate all the points raised during the rebuttal in our camera-ready version.
>
> We completely agree with you that the end-goal of whole XAI research is to incorporate humans in the loop, for providing them essential information to understand models. This specific work was focused on extracting model-faithful explanations, but our follow-up work will include humans since we plan to extend this work for debiasing models. We will incorporate this as our limitation/future-work as discussed in the rebuttal. We want to thank the reviewer again for your suggestion.

---

### Official Review · Reviewer_DAoS · 2025-07-01

**Clarity:** 3
**Significance:** 2
**Originality:** 2
**Rating:** 4
**Confidence:** 4

**Summary:**

The paper introduces a novel method for generating concept-based explanations for deep neural networks. The core of FACE is the integration of a KL divergence regularization term into the Non-negative Matrix Factorization (NMF) framework. This addition is designed to ensure that the extracted concepts are not only interpretable but also faithful to the model's decision-making process. The authors provide a compelling case for their approach, supported by theoretical guarantees and extensive empirical evaluation.

**Questions:**

1) FACE requires backpropagation through the classifier head to compute the KL divergence, which can be computationally intensive, especially for large models or datasets. Although the paper claims modest compute requirements, suitable for consumer-grade GPUs, this may still pose challenges in resource-constrained environments or real-time applications. A detailed analysis of computational complexity, including runtime and memory usage, would strengthen the paper’s claims.

2) The method assumes a model architecture with a clear separation between an encoder and a classifier head, as seen in ResNet34 and MobileNetV2. Its applicability to other architectures, such as transformers or models with integrated components, is unclear. This limitation restricts FACE’s generalizability across diverse deep learning models, which is a significant concern given the variety of architectures used in practice.

Overall this paper makes an incremental contribution to concept-based explainability by addressing the important problem of faithfulness in automatic concept extraction. The theoretical foundation is sound, the experimental evaluation is comprehensive, and the results clearly demonstrate the method's effectiveness. However, the work suffers from insufficient analysis of failure modes, and significant hyperparameter sensitivity that may limit practical adoption. Hence, I believe the paper would benefit from: (1) more thorough analysis of computational complexity and failure cases, (2) evaluation on a broader range of architectures and domains, (3) human evaluation studies, and (4) development of adaptive hyperparameter selection strategies.

**Ethical Concerns:**

["NO or VERY MINOR ethics concerns only"]

**Final Justification:**

I will keep my score for all the reasons that I already provided to the authors.

**Limitations:**

Yes

**Quality:**

3

**Strengths And Weaknesses:**

The primary contribution of FACE is its ability to ensure that these concepts are not only interpretable but also faithful to the model’s actual decision-making process, addressing a critical limitation in existing methods where extracted concepts may misrepresent the model’s reasoning. By incorporating a KL divergence term that minimizes the difference between the model's predictions on original and reconstructed data, FACE ensures that the discovered concepts are aligned with the model's predictive behavior.

The empirical evaluation of FACE is both thorough and convincing. The authors test their method on three diverse datasets (ImageNet, COCO, and CelebA) and two different model architectures (ResNet-34 and MobileNetV2), demonstrating the generalizability of their approach. The inclusion of multiple metrics to evaluate the quality of the matrix factorization, the faithfulness of the explanations, and the complexity of the explanations provides a comprehensive view of FACE's performance. The ablation studies on the regularization strength and the matrix decomposition rank further strengthen the paper by showing the impact of these hyperparameters on the results.

Weaknesses : 1) While the application is novel, the core technical contribution is relatively incremental - essentially adding a KL divergence term to standard NMF optimization. The theoretical analysis, while correct, relies on well-established inequalities (Pinsker's inequality) rather than developing new theoretical insights.

2) The strong alignment with the model’s predictions may lead to concepts that are overly specific to the particular model and dataset used. This could reduce the generalizability of the concepts to other models or domains, limiting their broader applicability. For example, if a model was trained with biases or on a specific data distribution, the extracted concepts might reflect those biases rather than generalizable features.

3) Since FACE relies on NMF, which is inherently linear and suited for extracting additive, parts-based representations, it may struggle to capture nonlinear or higher-order concepts (e.g., “smiling” in face recognition, which involves interactions between multiple facial features). While the classifier head might learn such interactions, the concepts themselves are limited to linear combinations, potentially missing complex relationships critical to the model’s reasoning.

4) FACE requires backpropagation through the classifier head to compute the KL divergence, which can be computationally intensive, especially for large models or datasets. Although the paper claims modest compute requirements, suitable for consumer-grade GPUs, this may still pose challenges in resource-constrained environments or real-time applications. A detailed analysis of computational complexity, including runtime and memory usage, would strengthen the paper’s claims.

---

> ### Author Rebuttal · Authors · 2025-07-30
>
> We want to thank the reviewer for their assessment. We appreciate that you understood that the faithfulness issue in concept-extraction is often neglected, and you found our approach and evaluation technically sound. Below we address some of your concerns:
> - **On incremental contribution:** We agree with the reviewer that the building blocks of FACE are classical. However, as the reviewer pointed out, the faithfulness issue in automatic concept extraction is overlooked, and our goal with FACE was to couple matrix decomposition with the classifier’s predictive behavior and make faithfulness an optimization target. We view FACE as a technique that can make an existing family of concept methods faithful to the underlying model.
>
> - **Concepts may reflect biases or specific data distribution used in training:** Yes, and this is intentional. FACE is a post‑hoc, model-faithful explainer: its goal is to reveal exactly what the given model relies on, including spurious or biased cues. This is crucial for auditing, diagnosing shortcut learning, and deciding whether to retrain or intervene on a model. Making concepts appear “more general” at the expense of fidelity would defeat that purpose. We actually ran a test with FACE on images with adversarial patches that force a target misclassification; FACE reliably isolates the patch as the top concept with high importance, exposing the attack vector. We can discuss this in the appendix.
>
> - **On linear/additive NMF:** We acknowledge that FACE, like most post-hoc XAI methods, uses a linear parts-based decomposition. However, with FACE, faithfulness is enforced at the model prediction level. So, if important nonlinear interactions were lost, the KL term would increase, and optimization would adjust the U and W matrices. High C-Del/C-Ins and low KL in practice indicate that a linear basis suffices locally to capture the model’s behavior.
>
> - **Computational cost and complexity:** FACE does not backpropagate through the whole network. The encoder g is frozen; gradients flow only through the low‑rank factors U and W and the (typically linear) classifier head h. Thus, each projected gradient optimization step costs a matrix product $UW^T$ plus a forward/backward pass through $h$. To address the reviewer's concern, we measured the runtime and peak memory on a single NVIDIA TITAN Xp (12 GB VRAM, CUDA 12.2). On 1,500 ImageNet images (model = Resnet34, rank = 25), preprocessing (computing activations + NNDSVD initialization) takes 5.6 ± 0.2 s, whereas the optimization takes 25.7 ± 0.2 s, with a peak memory usage of 3.55 ± 0.07 GB. For resource‑constrained scenarios, we note that FACE can be run on class subsets, with a smaller rank, or fewer steps (our loss plateaus early). We can discuss these computational analyses in the appendix.
>
> - **Applicability to ViTs**: FACE, like other post-hoc concept extraction, assumes a split f=h∘g. Directly applying FACE “as it is” to transformers like ViT is non-trivial, as ViTs lack fixed spatial feature maps, and “head” is often deeper/nonlinear. However, we see this as promising future work and will make this limitation explicit.
>
> - **Failure modes**: In Section 4.3, we discussed a failure case of FACE when an excessively large lambda value is used in high-class datasets like Imagenet and COCO, which degrades both faithfulness and accuracy. In addition, a very small rank can also underfit concepts. Second, as the reviewer pointed out, FACE is currently suitable for architectures where the split of the network into two functions is possible. We can extend the existing limitation by incorporating these points. Also, we have acknowledged lambda sensitivity (hyperparameter of FACE) as a limitation and leave fully automatic selection for future work. However, our current work still provides extensive sweeps and shows broad stable regions.
>
> - **Human evaluation**: The core problem FACE tackles is faithfulness, precisely because human plausibility can diverge from what the model actually optimizes. Therefore, we prioritized model-grounded perturbation tests over human judgment. Human studies are more appropriate for concept-bottleneck or user-facing systems where concepts are meant to match human notions; FACE targets alignment with the model’s internal reasoning. Hence, we refrained from human evaluation in this work. However, we can add this as our limitation.

---

> > ### Comment · Reviewer_DAoS · 2025-08-07
> >
> > I would like to thank the authors for their comprehensive response, that clarified most of my concerns. I have some minor comments:
> > 1. The intentional focus on model-specific cues, including biases, is a strength for auditing and debugging, and the adversarial patch example is a powerful illustration of FACE’s utility. However, this raises an ethical concern: if FACE exposes biased or spurious reasoning (e.g., relying on background elements), non-expert users might misinterpret these concepts as valid features without guidance. The authors should add a discussion on how to communicate such findings effectively, perhaps including the adversarial patch example as you suggest.
> >
> > 2. The discussion of λ and rank-related failure modes in Section 4.3 is adequate, and your acknowledgment of λ sensitivity is appropriate. However, additional failure modes, such as sensitivity to noisy inputs or model instability, remain unaddressed. For example, if the classifier head’s predictions are unreliable due to overfitting or adversarial perturbations, FACE’s reliance on these predictions could compromise concept quality.

---

> > > ### Author Response · Authors · 2025-08-07
> > >
> > > We thank the reviewer for their positive feedback.
> > >
> > > You are absolutely right that interpretability can also be misused in terms of misinterpretation or justification of flawed models. We have added a broader impact discussing this in Appendix A. We will expand this section based on your suggestion in the camera-ready version. Regarding the concept quality, we reiterate that regardless of how the model is trained, FACE yields faithful explanations meaning features used by the model in making its prediction. However, we will include your suggested additional failure models concerning noisy inputs or, instability of the model itself in the Limitation section. We sincerely thank the reviewer for these suggestions.

---

### Official Review · Reviewer_yEUV · 2025-07-03

**Clarity:** 4
**Significance:** 3
**Originality:** 3
**Rating:** 4
**Confidence:** 4

**Summary:**

In this work the authors propose FACE: a variant of NMF that incorporates a KL penalty on classification distributions to encourage reconstructions that also preserve downstream performance. The authors argue that this regularization encourages a more faithful concept extraction that aligns with the behavior of the model, and in the work demonstrate that this is the case through experiments while validating FACE’s competitive performance on traditional concept learning metrics such as sparsity and MSE.

**Questions:**

* In your convergence proof, Appx C Eqn (8), I think you dropped the Lipschitz Smoothness term. Then you could also explain exactly how sufficiently small $\nabla$ must be depending on the Lipschitz smoothness of your loss.
 * Do you want your sparsity metric to be high or low? In the table you want it to be high but in the Appendix based on your sparsity definition seems like you want it to be low as you are summing L0 norms.I’m also not super familiar with the Gini metric you are using, so I would make it extremely clear how the sparsity metrics are computed and thus if you want them higher or lower.
 * Also, why would increasing rank improve sparsity? I thought with NMF as you increase rank your decompositions are over more vectors, i.e. a less-sparse decomposition.
 * See weaknesses.

I am very willing to improve my score, but I think some of the above changes should be considered to improve the paper.

**Ethical Concerns:**

["NO or VERY MINOR ethics concerns only"]

**Final Justification:**

The authors satisfiably answered all my questions.

**Limitations:**

This is limited to models with a downstream task, which does not transfer well to generative systems.

**Quality:**

3

**Strengths And Weaknesses:**

## Strengths:
 * I really like the concept deletion and insertion metrics as a good causal test of the faithfulness of your explanation. I think these are the most important experiments in the paper. Is it possible to visualize the change in the image as you extract and delete concepts? Or show which concepts are being extracted and deleted in order? I would find that very interesting, at least in an example in the appendix.
 * The experiments validate the efficacy of the method.
 * I like the example of the gray hair and head vs face qualitative example. I would be interested to see if you could fix this “spurious correlation” in the model (for example train on young and old people with gray hair, assuming their faces look very different) and reapply FACE to see if the model is corrected properly.
 * I think the faithfulness of concept explanations is pretty under explored and I liked the methodology used by the authors to test this.
## Weaknesses:
 * Your proposition is just a direct statement of Pinsker’s inequality. I think the interesting component in your analysis is pointing out via Taylor approximation that even with small reconstruction, we have no bound on the change in logits. I might rewrite the proof to highlight this or drop it as a theoretical statement, similar to how you simply cite Pinkser’s in line 213. It would be interesting to provide a converse result or example where the gradient h(UW^T) is large despite the reconstruction being small.
 * As part of this your conclusion/implication of the proof is not really saying anything: if the KL is bounded, of course the model outputs are bounded, even without converting to TV.
 * Overall, I would rewrite 3.3 and 3.4 to highlight the lack of control we have over model outputs via the taylor expansion; then in 3.4 you can focus on the linear case and point out even in this case the jacobian of the softmax is out of our control and maybe you can construct a counterexample here.
 * Also, given you are modeling your classifier head as linear, you might be able to say something stronger regarding the lipschitzness/convergence of your algorithm.
 * Figure 4 doesn’t actually tell me that the KL divergence penalty is improving faithfulness: you say the metrics increase to $\lambda = 10^{-5}$ but to me the metrics appear flat. What happens when $\lambda = 0$? Figure 4 indicates to me that these metrics would not change, which is in disagreement with Table 2 assuming CRAFT is equivalent to $\lambda = 0$.

---

> ### Author Rebuttal · Authors · 2025-07-30
>
> We would like to thank the reviewer for their thoughtful review of our work, highlighting the importance of faithfulness-driven explanations and, efficacy of our evaluation. Below, we address your concerns:
>
> - **“Your proposition is just Pinsker’s inequality … the interesting bit is the Taylor part.”** In our work, Pinsker is not presented as a new theorem but as a bridge between our optimization objective (KL on predictive distributions) and a familiar notion of output deviation. The novelty is the placement of this regularizer inside NMF for concept extraction and the empirical demonstration that it improves faithfulness of explanations. We agree that the proposition as currently written can read like “KL small ⇒ prediction-difference small”, which is tautological; and as you have mentioned, “If KL is bounded, of course, the outputs are bounded”. We agree with you that the interesting part is in the first‑order Taylor expansion, which shows that small reconstruction error ∥$A−UW^⊤$∥ does not control the logits because $h$ can have a large Jacobian.  This is exactly why we add the KL term: *to control prediction difference directly*.
> In the camera‑ready, we will trim the Pinsker statement and move the emphasis to the Taylor observation (“reconstruction alone gives no bound on logits”) and also include converse examples as you suggested, such as: consider a linear head h(z)= $\alpha$$W$$z$ where $W$ is the learned weight and $\alpha$ is scaling factor on the logits learned implicitly during model training via weight-norm, batch-norm layers (or, equivalent to temperature of the softmax). Assume that the NMF factorization has made the Frobenius norm small between original A and A’ such that \delta = $||A’-A||_2$. Now, the logit change = $h(A’)-h(A)$ = $\alpha$ $W$$\delta$. Because $W$ is fixed, the product $\alpha$$W$ can easily amplify the logit change, changing the model prediction.
>
>
> - **Convergence proof: missing Lipschitz term & stepsize condition:** We thank the reviewer for suggesting to extend our existing proof. We have shared the updated version of the convergence underneath:
>
> Recall our optimization problem for FACE:
> L(U, W) = (1/2) * ||A - U Wᵗ||_F² + λ * KL( h(A) || h(U Wᵗ) )      (C.1)
>
> subject to *element-wise* non-negativity U ≥ 0,   W ≥ 0. We optimize this using projected gradient descent with a shared stepsize η > 0:
> Uₖ₊₁ = Proj⁺( Uₖ − η ∇ᵤ L(Uₖ, Wₖ) ) ; and Wₖ₊₁ = Proj⁺( Wₖ − η ∇𝓌 L(Uₖ, Wₖ) ).
>
> Let us stack the decision variables into Z = [U; W], and define the feasible set: Θ = { (U, W) | U ≥ 0, W ≥ 0 }.
>
> Assume the loss L(Z) is L∇-smooth over bounded subsets of Θ, i.e.,
> || ∇L(Zₖ) − ∇L(Zₖ₊₁) ||_F ≤ L∇ * || Zₖ − Zₖ₊₁ ||_F            (C.4)
>
> (The quadratic reconstruction error plus the smooth affine‑softmax KL term satisfies this.)
>
> By the standard descent lemma, for any Zₖ, Zₖ₊₁ ∈ Θ:
>
> L(Zₖ₊₁) ≤ L(Zₖ) + ⟨∇L(Zₖ), Zₖ₊₁ − Zₖ⟩ + (L∇/2) * ||Zₖ₊₁ − Zₖ||_F²     (C.5)
>
> Using non-expansiveness of projection, we get:
>
> ||Zₖ₊₁ − Zₖ||_F ≤ η * ||∇L(Zₖ)||_F                             (C.6)
>
> Inserting (C.6) into (C.5), we obtain:
>
> L(Zₖ₊₁) ≤ L(Zₖ) − (η − (L∇/2) * η²) * ||∇L(Zₖ)||_F²            (C.7)
>
> Thus, the loss decreases whenever the following condition holds:
>
> η − (L∇/2) * η² > 0    ⇨    0 < η < 2 / L∇
>
> So, for any stepsize:  0 < η < 2 / L∇, the loss sequence { L(Zₖ) } is monotonically non-increasing, bounded below, and convergent.
>
> - **Why would increasing rank improve sparsity?:**  Explanation sparsity (C‑Gini) measures how uneven the importance scores are across the $r$ concepts. A higher Gini means a few concepts dominate, hence a sparser explanation. C‑Gini is computed on the concept importance scores. When $r$ is tiny (few concepts), each concept must lump together many patterns, so their importance is forced to be similar, resulting in low Gini (flat distribution). As $r$ grows, concepts specialize: a handful become truly predictive for the class, while many receive near‑zero importance. The distribution of concept importance scores for the concepts becomes more skewed, so Gini rises. Hence, “rank ↑ ⇒ explanation sparsity ↑”.
>
> - **Sparsity metrics (U‑Spr vs. C‑Gini) and arrows:** We thank the reviewer for catching the confusion. Our current definition of U-Spr does measure the active fraction (density), so lower is better for sparsity. Because our projected gradient descent updates with non‑negativity rarely drive entries exactly to zero, FACE ends up with many very small but non-zero coefficients, yielding higher density than ICE/CRAFT, whose multiplicative updates often drive many entries to zero, so they look “sparser”. However, this property measures the matrix factorization quality; hence, there is no preference for low/high encoding sparsity. For camera-ready, we can either report density explicitly as U-Den (downarrow) or report sparsity as 1-U_Spr. This correction does not change our final conclusion: *FACE still achieves the lowest KL (prediction consistency) and strongest C-Del/C-Ins faithfulness, which is our motivation for FACE*. Our goal is faithful, succinct explanations, not necessarily the sparsest latent code.
> C‑Gini (explanation sparsity) is computed on the importance vector across concepts, not on U. For example, given a rank of 25, we obtain 25 different concept activation vectors (representing each concept), each with a concept importance score computed using the Sobol-index. Then, C-Gini measures the imbalance in the importance scores assigned to these 25 concepts. The goal in XAI, motivated by comprehensibility of explanations, is that sparser explanations are easily comprehensible. Meaning, if there is one or two highly important concepts among the 25 concepts, then it's much easier to understand how the model made predictions, rather than if the score is distributed evenly across all concepts. C-Gini is inspired by the inequality in the distribution of scores. So, in summary,  higher Gini ⇒ a few concepts dominate ⇒ higher sparsity. This is the sparsity we care about for interpretability.
>
> - **Figure 4 doesn’t show KL helps; what about λ=0? Is CRAFT = λ=0?:**  We have performed evaluation results with $\lambda$=0 (i.e., reconstruction‑only PGD‑NMF), where we observe a noticeable drop in faithfulness. On 10K ImageNet images (ResNet‑34), we obtain: C‑Ins=0.951, C‑Del​=0.796, C‑Gini=0.848. So the KL term does buy additional faithfulness. The apparent “flatness” in Fig. 4 stems from the saturation region: for small $\lambda$ (10⁻¹³–10⁻⁵), the KL term is strong enough to keep predictions aligned, so metrics plateau near their maxima. Finally, one point to note is that: CRAFT ≠ $\lambda$ =0. CRAFT uses multiplicative updates on spatial maps and pure MSE; our $\lambda$=0 uses projected gradient descent on averaged features. Empirically, their KL and C‑Ins/C‑Del differ (Table 2).
>
> - **Finding shortcut and “fixing” the model:** Thank you for this insightful idea. FACE is intentionally diagnostic: it is built to surface exactly which (possibly spurious) cues the model relies on so that practitioners can decide how to intervene. Repairing the model is orthogonal to (but downstream application of) explanation. Concretely, once FACE highlights a shortcut concept (e.g., a specific facial region for “gray hair” or an adversarial patch), one can employ data-level fixes such as upweight / resample examples, or employ loss-level fixes where we penalize reliance on the offending concept (e.g., add a concept-dropout or gradient regularizer on the FACE-identified dimensions). We are actively exploring these “concept-level debiasing” strategies and will mention this as a promising direction for future work.

---

> > ### Comment · Reviewer_yEUV · 2025-08-04
> > **Rebuttal Response**
> >
> > Thank you for your responses to my questions. I will increase my score, but it would help to get an understanding of how you will reframe proposition 1.

---

> > > ### Author Response · Authors · 2025-08-05
> > >
> > > We sincerely thank you for the positive feedback and for raising your score. We are happy to clarify how we plan to reframe Proposition 1.
> > >
> > > We will start with the motivation for KL regularization by explaining why reconstruction error is not enough. As you suggested, this will immediately set up the Taylor expansion argument as the central point, showing how a small reconstruction error can be amplified by the model's Jacobian. As discussed in the rebuttal, we will include a simple, clear example to demonstrate this. We will then present the KL term as the logical solution and frame Pinsker's inequality as a final implication that quantifies the benefit of our approach.
> > >
> > > We hope this plan clarifies our intentions and welcome any further suggestions you may have.

---

### Official Review · Reviewer_KKi1 · 2025-07-03

**Clarity:** 4
**Significance:** 4
**Originality:** 4
**Rating:** 5
**Confidence:** 4

**Summary:**

This paper presents FACE (Faithful Automatic Concept Extraction), a novel concept-based explanation method for deep neural networks. FACE builds on Non-negative Matrix Factorization (NMF) but introduces a KL-divergence-based regularization term to align concept-based explanations with the model’s actual predictive behavior. By incorporating classifier supervision, the method ensures that reconstructed representations from the learned concepts not only approximate encoder activations but also preserve the classifier’s output distribution. The paper provides theoretical justification that minimizing KL divergence bounds prediction deviation and promotes local linearity in the concept space. Empirical evaluations across ImageNet, COCO, and CelebA datasets show that FACE achieves higher faithfulness and explanation sparsity than state-of-the-art methods (e.g., CRAFT and ICE), while remaining competitive in reconstruction quality.

**Questions:**

Most of my concerns and questions are addressed in the Weaknesses section.

**Ethical Concerns:**

["NO or VERY MINOR ethics concerns only"]

**Final Justification:**

The authors have adequately addressed the concerns I previously raised, and the remaining issues appear to be minor and can be resolved in the camera-ready version. Therefore, I recommend this paper for acceptance.

**Limitations:**

The authors adequately address the limitations and potential negative societal impact of this work.

**Paper Formatting Concerns:**

No major formatting issues in this paper.

**Quality:**

3

**Strengths And Weaknesses:**

Strengths.
1. The proposed methodology is well-grounded, supported by a clear mathematical formulation and rigorous theoretical justification.
2. The empirical evaluation is comprehensive, demonstrating the effectiveness of the approach across multiple datasets (ImageNet, COCO, CelebA) and model architectures (ResNet34, MobileNetV2).
3. This work offers valuable insights into the nature of faithful model interpretation, and the use of KL divergence regularization, especially with its theoretical connection to prediction alignment via total variation bounds, is a particularly notable contribution. The method consistently outperforms existing baselines on key metrics such as faithfulness (C-Ins, C-Del) and sparsity (C-Gini).
4. The use of Concept Insertion (C-Ins) and Concept Deletion (C-Del) to quantitatively assess the impact of concepts on model predictions is another strength. While related perturbation-based metrics have been explored in prior work, their adaptation to the concept-based explanation setting and direct application to model performance evaluation represents a meaningful contribution.

Weakness.
1. In Figure 3 and lines 258–259, the authors suggest that FACE reveals the model's reliance on facial features for the "gray hair" class. However, if the model is leveraging spurious cues (i.e., shortcuts), such as facial features unrelated to hair color, one would expect the prediction to change when these facial features are altered. It would be helpful to investigate whether such changes in facial features lead to prediction shifts. If confirmed, this would further highlight the effectiveness of FACE in faithfully capturing the model’s true decision-making behavior, distinguishing it from baselines that may align more with human intuition than actual model reasoning.

2. In Proposition 1, the parameter ε is critical to the theoretical claim that KL divergence regularization guarantees predictive faithfulness and local linearity. However, the paper could benefit from a more detailed discussion of how ε behaves in practice. For example, is ε explicitly bounded in real-world applications? If not, how can one reliably assert that the L1 distance between predictive distributions (‖p − q‖₁) is small? Further clarification on how ε is interpreted, estimated, or controlled during training would strengthen the theoretical and practical understanding of the proposed method.

---

> ### Author Rebuttal · Authors · 2025-07-30
>
> We want to thank the reviewer for their thoughtful and positive assessment of FACE. We appreciate your recognition of our theoretical grounding, empirical breadth, and extensive evaluation. Below, we address the main concerns you raised:
>
> - **On spurious cues leading to change in prediction shifts**: Our goal is precisely to extract such concepts that the model relies on, even if those cues appear spurious for humans. This is the goal of faithfulness-aligned concept extraction method proposed with FACE. And, we already probe whether changing concepts result in prediction shift with our concept deletion (C-Del) test: *when we zero only the FACE-identified top concepts, we measure how the model’s accuracy changes*. Suggested by Reviewer FuWA, we also measured the changes in the logit of the original label by modifying the concepts and computing the logit-based insertion/deletion AUC on 10K ImageNet images (ResNet34). FACE was still able to obtain the highest scores compared to NMF baselines, as shown below.
> | Metrics/Method    | FACE | CRAFT | ICE |
> | -------- | ------- | ------- | ------- |
> | C-Ins  | 0.9466    | 0.8386 | 0.6403 |
> | C-Del | 0.9472     | 0.8413 | 0.6448 |
>
> To further emphasize FACE’s ability to expose shortcut behavior, we have run FACE on multiple images attacked with adversarial patches that force a targeted misclassification. FACE consistently isolates the adversarial patch as the top concept with the highest concept importance score. If required, we can include these qualitative experiments in the Appendix.
>
> - **Clarification on how $\epsilon$ is interpreted from Prop. 1:** In Prop. 1, $\epsilon$ is not a fixed constant; it is simply the empirical value of KL(p||q) measured between model prediction on the original and reconstructed activation vector. We do not assume a priori bound on $\epsilon$, but our hyperparameter $\lambda$ controls how small $\epsilon$ becomes. To make this explicit, we swept $\lambda$ over different magnitude on 10k ImageNet images using ResNet34 and report the mean KL and $||p-q|_1$ underneath:
>
>
>
> | $\lambda$ | KL (mean) | $\|\|p-q\|\|_1$ (mean) | √(2·KL) |
> |--------------|---------------|----------------------|-------------|
> | 1.00E-15     | 0.302         | 0.532                | 0.777       |
> | 1.00E-10     | 0.302         | 0.532                | 0.777       |
> | 1.00E-05     | 0.302         | 0.532                | 0.777       |
> | 1.00E-01     | 0.420         | 0.713                | 0.916       |
> | 1.00E+00     | 0.476         | 0.781                | 0.976       |
> | 1.00E+01     | 0.466         | 0.769                | 0.966       |
> | 1.00E+05     | 1.744         | 1.545                | 1.868       |
> | 1.00E+10     | 1.866         | 1.579                | 1.932       |
> | 1.00E+15     | 1.867         | 1.580                | 1.932       |
>
>
> For $\lambda$ used in our main experiment (moderate values), KL is small and the empirical ||p-q||_1 is well within Pinsker’s bound. Extremely large $\lambda$ hurts both KL and $||p-q||$ as already discussed in our ablation experiments in the main paper. In practice, we monitor KL during optimization and select $\lambda$ from the large “flat” region where both KL and ∥p−q∥  remain small. We can clarify this in the paper if required.

---

> > ### Comment · Reviewer_KKi1 · 2025-08-07
> >
> > Thank you for the clarifications and for carefully addressing my comments and running additional experiments. All of my concerns are addressed, I will keep my score as 5.

---

### Decision · Program_Chairs · 2025-09-17

**Decision:**

Accept (poster)

**Comment:**

The paper proposes a method for concept discovery specifically for non-negative matrix factorization, with one additional constraint that encourages similarity between classification outputs between original activation matrix and the reconstructed one. The method is very simple and the paper performed various analyses and proposed metrics like del/insertion -- similar to what has been used for counterfactual analysis in the literature. The reviewers made unanimous recommendations and the AC confirms it.